



# Spin-up Characteristics with Three Types of Initial Fields and the Restart Effects on the Forecast Accuracy in GRAPES Global Forecast System

Zhanshan Ma[1,2,3], Chuanfeng Zhao[1*], Jiandong Gong[2,3], Jin Zhang[2,3], Zhe Li[2,3], Jian Sun[2,3], Yongzhu Liu[2,3], Jiong Chen[2,3], Qingu Jiang[2,3]

[1]State Key Laboratory of Earth Surface Processes and Resource Ecology, and College of Global Change and Earth System Science, and Joint Center for Global Change Studies, Beijing Normal University, Beijing, 100875, China
[2]National Meteorological Center, Beijing, 100081, China
[3] Numerical Weather Prediction Center of China Meteorological Administration, Beijing, 100081, China

*Correspondence to*: Chuanfeng Zhao (czhao@bnu.edu.cn)

**Abstract.** The spin-up refers to the dynamic and thermal adjustments at the initial stage of numerical integration to reach a statistical equilibrium state. The analyses on the characteristics and effects of spin-ups are of great significance for optimizing the initial field of the model and improving its forecast skills. In this paper, three different initial fields are used in the experiments: the analysis field of four-dimensional variational (4D-VAR) assimilation, the 3-hour prediction field in the operational forecasting system, and the Final (FNL) Operational Global Analysis data provided by National Centers for Environmental Prediction (NCEP). Then, the characteristics of spin-ups in the GRAPES (Global/Regional Assimilation and Prediction System) global forecast system (GRAPES_GFS) under different initial fields are compared and analyzed. In addition, the influence of the lost cloud-field information on the spin-up and forecast results of the GRAPES model in the current operation is discussed as well. The results are as follows. With any initial field, the spin-up of GRAPES_GFS has to go through two stages - the dramatic adjustment in the first half hour of integration and the slow dynamic and thermal adjustments afterwards. The spin-up in GRAPES_GFS lasts for at least 6 hours, and the adjustment is gradually completed from lower to upper layers in the model. Therefore, in the evaluation of the GRAPES_GFS, the forecast results in the first 6 hours should be avoided. And the GRAPES_GFS with its own analysis field performs better than the one using FNL reanalysis data for the cold start in the spin-up, because the variation amplitudes of the temperature and humidity tendency are smaller and the spin-up time is slightly shorter. Based on the 4D-VAR assimilation analysis field, the forecast in the operational model is artificially interrupted and restarted after 3 hours of integration. In this process, as the cloud-field information is not retained, the spin-up should repeat in the model. The characteristics of spin-up are mostly consistent with those using the 4D-VAR assimilation analysis field as the initial field. However, as the cloud-field information is not retained in the current operation, the hydrometeor content in the atmosphere at the early stage of the forecast is underestimated, affecting the calculation accuracy of the radiation and causing a systematic positive bias of temperature and





geopotential height fields at 500 hPa. Besides, the precipitation is also underestimated at the early stage of the simulation, affecting the forecast of typhoon tracks.

## 1 Introduction

Norwegian scholar Bjerknes (1904) first explicitly proposed the theory of numerical forecasting in the early 20th century. After more than a century of development, it has become an effective way for studying climate change and its causes, as well as forecasting climate and weather. Besides, higher requirements have been also raised for the improvement of numerical forecast accuracy (Peter et al., 2015; IPCC 2013).

The numerical forecast accuracy is determined by a variety of factors. The European Centre for Medium-Range Weather
Forecasts (ECMWF) concluded that the steady improvement of the numerical forecast in the past 30 years can be mainly attributed to the improvement of the forecast model itself, the application of more observation data, and the development of data assimilation technology (Linus & Erland, 2013). Among them, the performance of the forecast model is mainly determined by the model resolution, the accuracy of finite difference methods, and the reasonability of the physical process parameterization schemes. Observation data mainly depends on the development of monitoring technology, especially the
application of satellite data. Data assimilation integrates observation data from different sources with model forecast elements so that the observation data can be comprehensively used by the models. The main purpose of data assimilation is to create a simulated atmosphere state closer to the real atmosphere, reduce the bias of the initial atmosphere condition, and thereby improve the quality of the initial field. In data assimilation, observation data from many sources are used. The uncertainties in the observation data, the inconsistencies among observation elements, and the model flaws (caused by model
dynamic assumptions, interactions between physical processes, static data initialization and the radiation balance adjustment, etc.) can lead to inconsistencies between the assimilated new observation input data and the original data in the model. Therefore, the model needs to readjust the dynamic and thermal processes at the initial stage of integration until a new statistical equilibrium state is reached. This process is called the spin-up in numerical model, and the time required to reach a new equilibrium state is called the spin-up time (Wolcott & Warner, 1981; Kasahara et al., 1992; Séférian et al., 2016; Sheng
et al., 2006; Liu et al., 2008; Xue et al., 2017). During the dynamic and thermal adjustment in the spin-up, spurious gravity waves could be triggered, causing a rapid increase in the root mean square error of the forecast variables in the model and an underestimation of the forecast precipitation (Wehbe et al., 2019; Qian et al., 2003). It leads to unreliable forecast results during the spin-up. Therefore, many studies generally do not consider the forecast results during the spin-up when evaluating the model forecasts (Lo et al., 2008; Kleczke et al., 2014; Xie et al., 2013; Zhao et al., 2012). If the spin-up time is too long
in the operational model, it would inevitably affect the forecast accuracy of the model. In addition, the overlong spin-up in the climate model or the ocean model can consume excessive computing resources (Duben et al., 2014). Therefore, studying the spin-up characteristics and reducing the spin-up time are of great significance for improving the model forecast and saving computing resources.





Due to different types and usages of numerical models, the spin-up time in different models is greatly different. For example,
in global climate models, glacial models, and ocean circulation models, the spin-ups usually take decades to hundreds of
years (Scher & Messori 2019; Danek et al., 2019; Rimac et al., 2017). But in a regional climate model or a land surface
model, only several weeks to several months are needed (Zhong et al., 2008; Rimac et al., 2017; Senatore et al., 2015; Giorgi
& Mearns 1999; Chen et al., 1997). In addition, the spin-up time is also affected by factors such as the simulation domain,
the simulation season, and the circulation intensity (Anthes et al., 1989; Errico et al., 1987). The spin-up time of short-term
weather forecast models is relatively short, usually several hours to about a dozen of hours (Weiss et al., 2008; Souto et al.,
2003; Kasahara et al., 1988). To reduce the impact of overlong spin-up on the accuracy of numerical forecasts, many
technical methods have been developed to shorten the spin-up time. For example, the "Distorted Physics", "Matrix-method",
"Jacobian-Free Newton-Krylov" are used in marine models (Bryan 1984; Khatiwala et al., 2005; Knoll & Keyes 2004). And
the cloud analysis method for assimilating unconventional observation data such as satellites and radars is used in the short-
term weather forecast model to improve the initial humidity field and cloud field, shorten the spin-up time, and improve the
short-term precipitation forecast (Li et al., 2018; Zhu et al., 2017; Xue et al., 2017; Li et al., 2011; Zhi et al., 2010; Xue et al.,
2003; Carlin et al., 2017).

The Global/Regional Assimilation Prediction System (GRAPES) is a numerical weather forecast model independently
developed by the China Meteorological Administration (CMA). It has become the core of the national numerical forecast
operational system in China. Numerical Weather Prediction Center of CMA has established a deterministic weather forecast
model system with a global horizontal resolution of 25 km and a national horizontal resolution of 3 km (Shen et al., 2017;
Zhang et al., 2019; Ma et al., 2018; Chen & Shen 2006). Hao et al. (2013) used the three-dimensional variational (3D-VAR)
system to perform the assimilation and analysis of initial fields in the GRAPES regional model, achieving a good forecast
result. The research by Zhu et al. (2017) showed that the cloud analysis method in the GRAPES regional model can
effectively shorten the spin-up time. After 1-hour integration in the model, the precipitation forecast is very close to the
observation, and it has a positive impact on the threat score of precipitation forecast within 12 hours. Li et al. (2011) also
showed similar findings. The assimilation module of GRAPES global forecast system (GRAPES_GFS) was upgraded from
3D-VAR to 4D-VAR assimilation system in June, 2018. The analysis and forecast ability of 4D-VAR assimilation system is
significantly better than 3D-VAR (Zhang L et al., 2019).  However, there are still many unknowns to be answered. For
example, what are the characteristics of the spin-up at the early stage of integration in GRAPES_GFS after the upgrade?
What advantages does it have, compared with the cold start simulation with the widely-used FNL (Final Operational Global
Analysis) reanalysis data provided by NCEP (National Centers for Environmental Prediction) (Kalnay et al., 1996)? In
addition, we should note that each forecast result of GRAPES_GFS is from the model integration forecast based on the 4D-
VAR assimilation analysis field 3 hours ago in the current operational forecast system. For example, the 1200 UTC forecast
result is based on the 4D-VAR assimilation analysis field at 0900 UTC. However, considering the habit of users (especially
forecasters) in using the forecast results, GRAPES_GFS integrates for 3 hours (to 1200 UTC) to retain the fields of basic
meteorological elements (U, V, T, Q, H, TS , Ps, etc.), and then the integration is terminated and restarts from 1200 UTC by





using the new-saved weather field data. The model forecast results thereafter are released, that is, the forecast results at 1200 UTC are obtained by users. In this process, the cloud-field information during the first 3 hours of integration is not retained

in the model, losing the cloud information formed after the 3-hour spin-up. Therefore, we need to fully diagnose and analyze the necessity of the repetition of GRAPES_GFS spin-up during the re-integration, and the impact of the lost cloud-field information on the later forecast. In this regard, the characteristics of spin-ups in GRAPES_GFS respectively using the 4D-VAR analysis data and the FNL data as the initial field are compared and analyzed, and the impacts of the cloud-field information loss in the current operation on the spin-up after the model restart and on later forecast results are discussed.

This paper aims to provide the scientific basis for understanding the characteristics of GRAPES_GFS at the initial stage of integration and improving the assimilation system and operational procedure.

The paper is organized as follows. In section 2, the GRAPES_GFS forecasting system and the experiment settings for one case study are introduced. In section 3, the main research results are presented. Finally, in section 4, the main conclusions are given, and some issues about spin-ups are discussed.

## 110    2 GRAPES_GFS and experiment setup

### 2.1 GRAPES_GFS

GRAPES is a global numerical weather prediction system that is composed of atmospheric model and variational data assimilation system (3D-VAR/4D-VAR). The framework of the atmospheric model is a fully compressible non-hydrostatic dynamical one with semi-implicit and semi-Lagrangian time difference scheme. In the horizontal direction, the equidistant

latitude-longitude grid system with the Arakawa-C grid and central difference of second order accuracy for variable staggering is used; and in the vertical direction, the height-based terrain-following coordinate with the Charney-Phillips staggering is adopted. Forecast variables of GRAPES_GFS include the dimensionless air pressure (Exner function), potential temperature, three-dimensional wind field components, and specific humidity. And it introduces the Piecewise Rational Method (PRM) scalars (Su et al., 2013) into the model, which is a scheme of water vapor advection. The physical

parameterization schemes used in the GRAPES_GFS operation mainly include the long-wave and short-wave radiation scheme (the rapid radiative transfer model (RRTMG)) (Morcrette et al., 2008; Pincus et al., 2003), the land surface scheme (the Common Land Model (CoLM)) (Dai et al., 2003), the planetary boundary layer scheme (Medium-Range Forecast (MRF)) (Hong & Pan 1996), the deep and shallow cumulus convection parameterization scheme (the New Simplified Arakawa-Schubert (NSAS)) (Arakawa & Schubert 1974; Liu et al., 2015; Pan & Wu 1995). The cloud physics scheme

includes the macro cloud scheme dealing with the condensation process under the unsaturated condition of grid-average water vapor, a double-moment cloud microphysical scheme, and a cloud cover prognostic scheme (Chen et al., 2007; Ma et al., 2018). The 4D-VAR is adopted in GRAPES_GFS (Zhang et al., 2019).





### 1.2 Experiment setup

In this paper, the weather process in GRAPES_GFS with the operational forecast time of 0000 UTC on August 9, 2019 is

taken as an example, and three experiments are set up to analyze the similarities and differences in the spin-up characteristics of the GRAPES_GFS model using different initial fields. The settings are shown in Table 1. In the first experiment, the analysis field provided by the 4D-VAR assimilation analysis system in the operational forecast at 2100 UTC on August 8, 2019 is used as the initial field to directly perform model integration forecasts, and the initial time is 2100 UTC on August 8. This experiment is called G21. In the second experiment, the initial field in the operation is used, and the results are the

forecast products provided by GRAPES_GFS operational system to the users. At 0000 UTC on August 9 (3 hours after the beginning of integration in G21), it retains the model variables required by the pre-processing system and stops the integration, but loses the cloud-field information (e.g. hydrometeors and cloud cover). And then the model restarts at 0000 UTC on August 9 with the reserved forecast-field information for forecasting. This experiment is called G00. The third experiment uses the initial field from the NCEP FNL reanalysis data at 0000 UTC on August 9, 2019 to perform integration

forecast. The purpose is to compare the spin-up characteristics of GRAPES_GFS model respectively using its own analysis field and FNL reanalysis field as the initial field. This experiment is called F00. To analyze the impacts of the initial field on the forecast, G00 and F00 produce a continuous 72-hour forecast. As G21 starts the integration 3 hours earlier than the other two, the forecast of G21 lasts for 75 hours to ensure the same forecast and analysis period with G21 and G00.

All the three experiments are based on the GRAPES_GFS operational model, with a horizontal resolution of 0.25°, 60

vertical layers, and a model integration time step of 300 s. The physical schemes used are from operational solutions (as described in section 2.1), and the assimilation module is 4D-VAR assimilation system. To explicitly analyze the spin-up characteristics of the GRAPES_GFS at the early stage of integration, the results of each integration step are outputted, and the temperature tendency (TT) and water vapor tendency (WVT) fields at each model layer during the dynamic and physical processes are retained.

In addition, the cloud-field information has not been saved during the restart in the current operation. To examine its impact on the accuracy of the later forecast, this study investigates the Super Typhoon "Lekima" (No. 1909) that landed on China during the selected forecast period, and the forecast differences in cloud, precipitation field and typhoon track during "Lekima" between G00 and G21 are analyzed.

## 3 Results

### 3.1 Characteristics of spin-ups

#### 3.1.1 Characteristics of total WVT and total TT

To analyze the spin-up characteristics of GRAPES_GFS, the initial fields in F00, G21, and G00 are used to perform the integration, and the temporal variations of the average total WVT and TT at different heights from 0000 UTC to 1200 UTC





are calculated, as shown in Fig. 1. Seen from the figure, both the WVT and TT show sharp fluctuations at the initial stage of

the integration in the three experiments, especially during the first hour. After 3–6 hours of spin-up adjustment, the variation

magnitudes of WVT and TT become gradually gentle, but the variation characteristics vary with different initial fields. At

the early stage of the integration, the WVT is adjusted in F00 and G21, with the amplitude of $-4.5$ g kg$^{-1}$ d$^{-1}$. In G21, the

water vapor adjustment occurs in the lower layers of the model (850 hPa and 925 hPa), while the WVT is relatively gentle

without an obvious adjustment in the upper and middle layers (500 hPa and 300 hPa). In F00, the water vapor adjustment

occurs at the upper levels of the model at the early stage of integration. The WVT at 300 hPa can reach $-4.5$ g kg$^{-1}$ d$^{-1}$, but

it weakens immediately afterwards, probably due to the supersaturated water vapor in the initial field from FNL data. In F00,

the WVT in the lower layers of the model is also significantly larger than that in G21. For example, at 850 hPa, the WVT in

F00 maintains about 1 g kg$^{-1}$ d$^{-1}$ for relatively long time, but that in G21 mostly changes within 0.5 g kg$^{-1}$ d$^{-1}$. The

corresponding temperature adjustment processes in the two experiments present the same variation characteristics as the

WVT adjustment. Therefore, the spin-up in the integration using the analysis field of GRAPES_GFS as the initial field is

gentler than that using the FNL reanalysis data as the initial field.

In G21 and G00, both the variations of WVT and TT are very consistent, indicating that G00 has well inherited the

temperature and humidity structure of G21. However, G00 still needs to go through the spin-up during which a gradually

stable adjustment process follows a sharp fluctuation at the early stage of integration, that is, the dynamic and thermal

adjustments are required to reach a statistical equilibrium state in the model. At the initial stage of integration in G00, the

variation amplitudes of WVT and TT are smaller than those in G21, but greater than those in G21 after the 3-hour integration.

It shows that although G00 can retain the temperature and humidity structure of G21, the loss of cloud-field information in

the operation still has a destructive effect on the model equilibrium state after 3-hour adjustments. Based on the variation of

TT, the spin-up time required for G00 is generally less than that for G21. It takes about 6 to 8 hours to reach a TT

equilibrium state in G21, but it is less than 6 hours in G00.

### 3.1.2 Tendency characteristics of the model dynamical and physical processes

Fig. 2 shows the temporal variation of mean WVT in the dynamic and physical processes at different heights in F00, G21

and G00. In the middle and upper layers of the model (Figs. 2a and 2d), there is a drastic adjustment in the atmosphere at the

early stage of the integration in F00. It may be due to the supersaturated water vapor in the initial field from FNL data, which

causes the cloud to condense very quickly, and thus a relatively stable state is reached, after three integration steps. At this

level in G21, the WVTs at the first few integration steps are slightly larger than that at the subsequent integration steps,

while the variation of water vapor is mainly caused by the co-action of cloud and convections. There is not much difference

in the dynamic field tendencies between G21 and F00. The magnitudes of the WVTs in the dynamic processes of the two

experiments are also very close: around 0.5 g kg$^{-1}$ d$^{-1}$ at 500 hPa and 0.25 g kg$^{-1}$ d$^{-1}$ at 300 hPa. Therefore, the differences

of the upper-middle-level water vapor adjustments in the spin-ups between G21 and F00 are mainly caused by physical





processes, and there is a good consistency in the dynamic process between the two experiments. At 925 hPa (the lower layer of the model), the total WVT stays around 1 g kg$^{-1}$ d$^{-1}$ in F00 after three integration steps, humidifying the atmosphere. In G21, it reaches a relatively stable state after six integration steps, and water vapor decreases overall. As the WVTs of the dynamical processes in F00 and G21 have the same magnitude around 0.25 g kg$^{-1}$ d$^{-1}$, the difference of the total WVT

between G21 and F00 is mainly caused by physical processes. The effect of the boundary layer on the WVT is similar in both experiments and the WVT is almost 3 g kg$^{-1}$ d$^{-1}$. The greatest difference between the two experiments is mainly caused by the convection scheme. The convection in F00 is relatively gentle, and the WVT from convection is around -1 g kg$^{-1}$ d$^{-1}$. In contrast, due to the strong dehumidification ability of convections in G21, the WVT is between -5 g kg$^{-1}$ d$^{-1}$ and -2.5 g kg$^{-1}$ d$^{-1}$, which is significantly stronger than that in F00. At 925 hPa, the water vapor mainly decreases due to the strong

convection process in G21. Such a significant difference in the convection processes between F00 and G21 may be related to the low-level temperature and humidity structures and the triggering conditions for convections. Meanwhile, it can be seen that the difference in the initial field of the model can significantly affect the physical process.

In summary, in the middle and upper atmosphere, the fluctuation of WVT in G21 is weaker than that in F00, indicating the advantage of using the data assimilation cycling as the initial field. Both experiments quickly reach a quasi-equilibrium state

after dramatic adjustments over several integration steps. The water vapor adjustment in spin-ups mainly occurs in the lower atmosphere of the model. The difference is mainly caused by different convection schemes. At the same time, different initial fields of the temperature and humidity structure may lead to a great difference in the dehumidification ability of convections. For G00 and G21, the WVTs of the dynamic and physical processes have roughly the same characteristics. At all of the three levels, the WVTs in G00 are slightly lower than those in G21.

In the middle and upper layers of the model, the dramatic change of the TT in F00 mainly occurs within the first half hour of the integration. The TT in the physical process is mainly caused by the water vapor condensation due to the cloud and convection processes (Fig. 2). Compared with the convection process, the cloud physical process can cause greater temperature adjustments. For example, at 500 hPa, the global average heating produced by the cloud microphysical condensation process at the initial time can exceed 5 K d$^{-1}$ and it takes four integration steps to reach a relatively stable state.

But at this level, the TT caused by the convection process is 3 K d$^{-1}$, and it only needs one integration step with the drastic adjustment to get relatively stable. In addition, the TT caused by the dynamic process fluctuates greatly at the first half hour of the integration. For example, at 300 hPa, the TT fluctuates between 1.1 K d$^{-1}$ and 1.5 K d$^{-1}$, and it requires extra 3 or 4 integration steps to reach a relatively stable state compared to the physical process. Nevertheless, after half an hour of severe fluctuations, the TT caused by dynamic and physical processes tends to be relatively stable. Overall, the temperature

increases by 0.25 K d$^{-1}$ to 0.5 K d$^{-1}$ in the middle and upper atmosphere in F00. Compared with that in middle and upper layers, the TT variation caused by the dynamic and physical processes in the lower layer of the model (Fig. 3g) shows a relatively large and rapid adjustment at the first integration step. But no drastic adjustment is shown afterwards, and its variation is relatively stable. The TT at 925 hPa in F00 is mainly caused by dehumidifying and heating of the atmosphere from the convection parameterization. The average global heating is between 1.5 K d$^{-1}$ and 2 K d$^{-1}$. The TTs caused by other





processes are negative. Overall, in F00 the atmospheric temperature is reduced at 925 hPa, with an amplitude of about -1.2 K d$^{-1}$.

In G21, the TT in the middle and upper layers also experiences a dramatic adjustment in the first half hour of the integration (Figs. 3b and 3e), and the main reason for the fluctuation is the dehumidification and heating in the convection process, which is different from that in F00 caused by the cloud physical process. The temperature increase caused by the convection

process in G21 is 1 K d$^{-1}$ to 2.5 K d$^{-1}$, which is about twice that in F00. The TT caused by the cloud physical process in G21 varies relatively gently. Similar to F00, the TT caused by the dynamic process in G21 also shows obvious fluctuations, which may be caused by the drastic variations of physical processes. In the lower layer of 925 hPa (Fig. 3h), the positive TT in G21 is also caused by convective dehumidification and heating, while other processes lead to cooling. In terms of the total TT (dynamic core plus physical processes), F00 has a cooling effect with a value of -1 K d$^{-1}$, while G21 has a warming

effect with a value within 1 K d$^{-1}$. The temperature increase rate of G21 gradually decreases with the integration step.

The characteristics of the TT variation in G00 are consistent with those in G21 (Figs. 3c, 3f and 3i). In the first half hour, it also has a drastic adjustment process, with the adjustment amplitude close to or slightly smaller than those in G21. After half an hour, the temperature tends to be relatively stable. The TT variation in G00 indicates that although G21 has undergone a 3-hour spin-up, G00 needs to undergo it again due to the loss of cloud-field information during the restart, and its fluctuation

amplitude is not substantially smaller than that of G21.

### 3.1.3 Evolution characteristics of the cloud field

The comprehensive adjustment effect of the dynamic and the physical processes on the water vapor and temperature in the numerical model can be presented by the cloud state. To reveal the dynamic and thermal adjustment processes in GRAPES_GFS system at the beginning of the integration and the time required for the analysis model to reach the statistical

equilibrium state (spin-up time), this section uses the total grid number of cloud (TGNC) in the model as the index for analyses. Although the cloud is changing locally, the total area covered by cloud can be regarded as a constant globally on average. Therefore, TGNC is used as the analysis index, and the model is considered to have completed the spin-up when the TGNC gets relatively stable. The total hydrometeors content (THC) greater than 1.0 e-4 g kg$^{-1}$ in GRAPES_GFS is defined as the grid with cloud, and the TGNC at a global scale or a certain height is the sum of all the grids in the corresponding

cloud area.

Fig. 4 shows the vertical distributions of TGNC at different lead time in three experiments. It can be seen that no matter the GRAPES_GFS model is cold-started with reanalysis data (F00, Fig. 4a) or warm-started with the 4D-VAR analysis field as the initial field (G21, Fig. 4b), the TGNC experiences rapid generation and growth during the 3 hours after the beginning of integration in the two experiments, especially in the middle- and low-cloud regions below 300 hPa. After 3 hours of

integration, the TGNC grows relatively slowly, while after 6 hours of integration, the TGNC basically gets stable. However, the time required for the TGNC to reach the equilibrium state is slightly different at different heights. In F00, the integration time required for the TGNC to gradually reach the statistical equilibrium state below 850 hPa is 6 hours. Note that the



statistical equilibrium state is defined when the difference between the TGNC after 24-hour integration below 850 hPa and the TGNC after 6-hour integration is insignificant. However, it takes 6–12 hours for the TGNC to get stable and completes

the spin-up above 850 hPa. For G21, the TGNC of the middle and low cloud below 300 hPa needs 6 hours to reach the statistical equilibrium state, while the TGNC of the high cloud above 300 hPa needs 6–12 hours. It can be seen that the GRAPES_GFS using the analysis field from its own data assimilation cycling enables the cloud field in middle and upper layers to reach the equilibrium state earlier than that using FNL data for the cold start. In addition, GRAPES_GFS is gradually adjusted from the lower to the upper layers of the model to reach the equilibrium state, which is consistent with the

evolution characteristics of the thermodynamic process in the troposphere. For the cloud above 500 hPa, the TGNC in F00 is significantly more than that in G21, which is related to a higher relative humidity of the initial field (Fig. 6).

In G00 (Fig. 4c), the growth of TGNC is found to be much slower than that in G21, especially the TGNC of the middle and upper cloud. For example, at time 3 hours after the beginning of G00, the TGNC of the middle cloud is mostly between 15 and 20, while the TGNC in G21 can reach 25–30. The reason may be that the humidity and temperature fields of the model

in G21 are already in a relative equilibrium state after 3-hour spin-up. Meanwhile, as the restart of GRAPES_GFS has lost the cloud-field information (light blue dotted line) from the 3-hour integration, the TGNC cannot reach the previous magnitude in the middle and upper layers even if it has been integrated for 24 hours in G00 (Fig. 4c, solid purple line).

Fig. 5 shows the distributions of THC at 400 hPa at different lead time in the three experiments. It can be seen that the temporal variation characteristics of THC and its horizontal distribution at 400 hPa have consistent results with those shown

in Fig. 4. In F00 and G21, as supersaturated water vapor is removed from the initial field, the cloud is quickly generated at the first integration step of the model. The THC rapidly increases within 1 hour, and the cloud area with high hydrometeor content is constantly expanding. For example, at time 1 hour after the integration in F00, the THC in most areas of the Pacific Warm Pool is 0.2 g kg$^{-1}$. With the further adjustment of the spin-up, the THC in this area gradually decreases, and maintains a relatively equilibrium state after 6 hours of integration. The variation characteristics of the THC in the storm

track area (60°S–30°S) in the southern hemisphere are similar to those in the warm pool area, but less significant.

Experiments using the 4D-VAR analysis field to provide the initial field (Figs. 5e–5h) show that the variation characteristics of THC at 400 hPa are generally consistent with those in F00. After the first integration step of GRAPES_GFS, cloud areas are quickly generated in tropical and mid-latitude areas. Due to the rapid development of convection processes in tropical areas, more cloud with THC of 0.0–0.05 g kg$^{-1}$ appears. After 3 hours of integration, the development of the cloud area

gradually weakens. After 6 hours of integration, the variations of the range and shape of the cloud area are no longer obvious, and it can be considered that a relatively equilibrium state is reached. From the view of absolute value of THC in the cloud area, although the difference in the distribution range of the cloud is insignificant, the THC in G21 is significantly less than that in F00 due to the different temperature and humidity conditions in their initial fields (Fig. 6).

Since G00 does not retain the cloud-field information after 3 hours of integration in G21 (the THC in Fig. 5g), the model

needs to undergo a new cloud-generation process when restarting the integration. However, as the dynamic and thermal fields are obtained after 3 hours of adjustments in G21, the relative humidity has undergone a condensation process, making



the atmosphere of G00 with much weaker supersaturation than that at initial time. Therefore, unlike F00 (Fig. 5a) or G21 (Fig. 5e), in which large-scale cloud appears instantaneously, the cloud field in G00 can only be gradually generated by the dynamic and physical processes of the model. It can be seen from Figs. 5i–5k that this process is relatively slow, and a

relatively stable cloud distribution does not appear until 3 hours after the integration. The cloud range in G00 at that time is smaller than that in G21, and it generally reaches the equilibrium state after 6 hours of integration. The influence of slower generation and smaller range of the cloud in G00 on the model forecast results will be analyzed and explained in section 3.2. To reveal the reason why the TGNC (Fig. 4) and the THC (Fig. 5) in the upper layers of the model in F00 are significantly higher than those in G21, the difference of water vapor content and relative humidity at 400 hPa is analyzed, and the results

are shown in Fig. 6. Fig. 6c shows that the specific humidity in the initial field of F00 is generally higher than that of G21 in the tropical areas and the middle and high latitude areas of northern hemisphere, especially in the tropical warm pool area where the difference is mostly over 0.2 g kg$^{-1}$. The relative humidity can reflect the degree of water vapor saturation. Fig. 6d shows that the relative humidity of the initial field from the FNL reanalysis data is relatively higher than that from the 4D-VAR analysis field in the tropical warm pool, Intertropical Convergence Zone (ITCZ), and middle and high latitude areas at

400 hPa. It means that the water vapor is more likely to get saturated using the FNL reanalysis data as initial field. Thus, the cloud area is larger and the THC is higher at the beginning of the integration. It is not difficult to conclude that there are differences in the structure of atmospheric temperature and humidity among different initial field data, which significantly impacts the spin-up characteristics of the model as well as the cloud formation and development. It also suggests that we need to pay more attention to the analysis quality of water vapor in data assimilation.

### 3.2 Impacts on later forecast results

It can be seen from section 3.1 that the cloud-field information formed in the first 3 hours of integration has not been saved in the operation, so the model must restart the spin-up, and THC appears to be significantly less in the new spin-up. In order to discuss the impact of the restarted spin-up and the less THC on the later forecasts by GRAPES_GFS, the global radiation field and synoptic situation field (temperature and geopotential height) are analyzed in this section. The cloud and

precipitation fields and the typhoon track of the super typhoon "Lekima" that landed on China during the simulation period will be analyzed as well.

### 3.2.1 Impacts on global radiation

Fig. 7 shows the zonal mean distributions of averaged column cloud water content (CCWC), the outgoing longwave (OLR) at the atmosphere top and the downward longwave at ground (GDLW) simulated by G21 and G00 from 0000 UTC to 0300

UTC on August 9, 2019, as well as the distributions of difference between them. It can be seen from Fig. 7a that the total zonal-averaged CCWC forecasted in G00 is systematically smaller than that forecasted in G21. The areas with smaller CCWC are mainly located in the Southern Hemisphere storm track, tropical low-latitude areas, as well as middle- and high-latitude areas in the northern hemisphere with active cloud. Among them, the area with the smallest CCWC is the active area





of Southern Hemisphere storm track, with the CCWC difference reaching 240 g m$^{-2}$, and there are also some areas with the
CCWC difference over 200 g m$^{-2}$ in the northern hemisphere. From the OLR and GDLW predicted in the two experiments, it can be seen that the OLR predicted in G00 is systematically larger than that in G21, with the maximum bias (20 W/m$^2$/s) appearing in the Southern Hemisphere storm track. This is due to the interaction between cloud and radiation, as well as the underestimation of the CCWC. In terms of GDLW, the reduced CCWC weakens the atmospheric warming effect, resulting in systematically smaller GDLW in G00 than in G21. In most areas, the GDLW is smaller than the observation by over 10 W
m$^{-2}$ s$^{-1}$, and the regions with the largest bias are the middle- and high-latitude areas of the Southern Hemisphere and high-latitude areas of the Northern Hemisphere.

### 3.2.2 Impacts on the global temperature and geopotential height fields

The change in the calculation of the radiation flux induced by cloud would seriously affect the atmospheric temperature field and geopotential height field. Fig. 8 shows the difference distributions of the 500-hPa temperature field and the geopotential
height field at two lead time between G00 and G21. It can be found that as there is less hydrometeor in the cloud in G00 than in G21, the temperature field in G00 at different forecast moments shows a systematic warming of more than 0.1 K in the tropical low-latitude and middle-high-latitude areas with active cloud. With the increase of the lead time, the warming area is expanding and the degree of warming gradually increases. For example, after 72 hours of integration, the warming in many areas is larger than 0.2 K, and it can reach 0.5 K in some areas. Systematic biases also appear in the corresponding
geopotential height field. Compared with those in G21, the geopotential height fields in G00 have also systematic positive biases. For example, in the first 24 hours of integration, the systematic biases in the geopotential height field are above 0.5 gpm, and the positive bias can exceed 1 gpm in areas with active cloud. After 72 hours of integration, the geopotential height field in the tropical area still shows a systematic positive bias, while in the middle- and high-latitude areas, the bias of the geopotential height field shows the structure with an alternation of positive biases and negative biases due to the biases of the
weather system location predicted in the two experiments, but in most areas the forecast fields are still higher than the observation.

### 3.2.3 Impacts on typhoon forecasts

This section analyzes the biases of the cloud field, precipitation field, and the track of the Super Typhoon "Lekima" (No. 1909) and Typhoon "Krosa" (No. 1910) in 2019 during the forecast period to evaluate the impact of the lost hydrometeor
information on typhoon forecast operation in GRAPES_GFS. During the forecast, "Lekima" and "Krosa" appear as double typhoons in the western Pacific. "Lekima" landed on North China from offshore areas, while "Krosa" continued to develop on ocean. G00 and G21 give the same forecasts for the cloud and precipitation fields of "Lekima" and "Krosa". Here, we only show the impact on the cloud and precipitation of "Lekima" by the lost hydrometeor information on typhoon forecast operation of GRAPES_GFS. In the last part, the path-forecast biases for the two typhoons are both given.





Fig. 9 shows the evolutions of the averaged CCWC and total content of ice water (TCIW) within the main cloud area of "Lekima" (117°E–130°E, 22°N–34°N) simulated in G00 and G21 from 0000 UTC on August 9 to 0000 UTC on August 10, 2019. It can be seen that the TCIW predicted in G00 at the early stage of integration is obviously underestimated. The averaged TCIW values in G21 are maintained within 850–1000 g m$^{-2}$ from 0000 UTC to 0900 UTC on August 9, while the TCIW is only 480 g m$^{-2}$ at the initial time of G00. G00 needs to restart the spin-up. During the spin-up, the TCIW predicted

in G00 increases rapidly, with the greatest strengthening during 0000 UTC to 0600 UTC. After 3 hours of the integration, the TCIW increases rapidly from 480 g m$^{-2}$ to 820 g m$^{-2}$. After 6 hours of integration, the TCIW is close to 900 g m$^{-2}$. In G00, the TCIW is not as large as that in G21 until 9 hours after the beginning of integration.

Fig. 10 shows the difference distributions of both 3-hour and 24-hour accumulated precipitation (since 0000 UTC August 8, 2019) of "Lekima" between forecasts of G00 and G21.The most significant difference of the 3-hour cumulated precipitation

appears within the first 3 hours of integration in G00. The 0000 UTC–0300 UTC precipitation forecasted in G00 presents a systematic underestimation when compared with G21, and the biases are all above 1 mm. The precipitation bias in the center of "Lekima" can even exceed 5 mm (Fig. 10a). As shown in Fig. 9, after 3 hours of adjustments, the total CWP and TCIW in the typhoon system in G00 grow rapidly and get close to the magnitudes of them in G21. Therefore, the difference of the 3-hour precipitation between forecasts of G00 and G21 is no longer significant during 0300 UTC–0600 UTC and 0600 UTC–

0900 UTC, and there is no more systematic bias (Figs. 10b and 10c). The phase differences of the weather system lead to the structure with an alternation of positive biases and negative biases for the precipitation difference.

It can be found from Fig. 10d that the lack of cloud-field information has a significant impact on the simulation of the accumulated precipitation in the first 24 hours of "Lekima". The negative biases dominate the central area of the typhoon, that is, there is an underestimation of precipitation with the maximum bias of 5–10 mm. While in the spiral cloud zone

around the typhoon, there is a structure with an alternation of positive and negative biases, which is related to the location bias of the weather system simulated in the two experiments in this area.

Fig. 11 shows the forecast track evolution of "Lekima" and "Krosa" in G00 and G21 within the lead time of 72 hours. Overall, G21 performs better than G00 in predicting the tracks of these two typhoons, and there are different characteristics for the track forecast biases of the two different typhoons. "Lekima" landed on the coast of Chengnan Town, Wenling City,

Zhejiang Province at 1545 UTC on August 9, 2019. There is not much difference in biases of the track forecast between G00 and G21 before "Lekima" landing. While the biases appear to be different after the landfall (1600 UTC), and the track forecast in G21 is slightly better than that in G20 around the landfall. After the landfall, the track biases change continuously during the 27th to 42th hour and 54th to 60th hour of the forecast, the track bias in G21 is smaller than that in G20. The maximum difference between the two track forecasts can reach 32 km. From the 65th to 72th hour, the forecast track bias in

G21 is slightly larger. For "Krosa", during the first 42 hours, the biases of the tracks forecasted in G00 and G21 are not much different. But the forecast tracks of the two become different after the 42th hour, with the track bias in G00 becoming larger. In most forecasts after the 42th hour, the track biases in G00 are over 20 km and larger than those in G21.



Overall, G21 performs better than G00 in the track forecasts of "Lichma" and "Krosa" within the lead time of 72 hours, especially in the forecast of "Krosa". For "Krosa", the forecast track on the ocean is less affected by other factors, so the

forecast track biases at the later stage of the forecast are significantly smaller. It shows that GRAPES_GFS performs better in continuous-integration forecasts, and the interruption in the operation is destructive to the typhoon track forecast.

## 4 Conclusions and discussion

To analyze the characteristics of the spin-up at the early stage of integration in GRAPES_GFS, this study adopted three different initial fields, namely the 4D-VAR analysis field (G21), the field obtained by interrupting and restarting the 4D-

VAR analysis field after 3 hours of integration (G00), and the field based on FNL reanalysis data for cold start (F00). Moreover, the differences between G00 and G21 on the later model forecast results were analyzed to evaluate the impact of current operational procedure on GRAPES_GFS forecasts. The main conclusions are as follows.

All the three different experiments using different initial fields show that the spin-up of GRAPES_GFS has to go through 2 stages: the dramatic adjustment in the initial half-hour of integration and the slow dynamic and thermal adjustment

afterwards. In the middle and lower layers of the model, the spin-up takes 6 hours to reach the equilibrium state, and takes longer in the upper layers. The dynamic and thermal adjustment is gradually completed from the lower to the upper layer of the model.

The GRAPES_GFS using its own analysis field as the initial field (G21) is gentler in the water vapor and temperature adjustment in the spin-up than the GRAPES_GFS using FNL reanalysis data for cold start (F00), and the time required is

slightly shorter. Due to the different structures of temperature and humidity in the two initial fields, the differences of physical processes in the model spin-up adjustment are obvious, especially the convections and cloud physical processes. However, the differences in dynamic processes are unobvious. G00 needs to repeat the spin-up. Its dynamic and thermal adjustments are similar to that in G21. The temperature and humidity adjustment in G00 is slightly weaker than that in G21, and its spin-up is slightly shorter.

In G00, the cloud-field information is not retained during the current operation of GRAPES_GFS. It shows that G00 significantly underestimates the atmospheric CCWC and TCIW at the early stage of forecast, which would affect the calculation accuracy of radiation and result in systematic positive biases in temperature and geopotential height fields at 500 hPa. Due to the lack of cloud-field information, the accumulated precipitation in the first 3 hours of integration in G00 is significantly underestimated. The 24-hour accumulated precipitation in the typhoon center is also less than that in G21, and a

destructive effect is made on the typhoon track forecast.

Regarding the influence of the lost cloud-field information in the GRAPES_GFS operation on the forecast results, this paper mainly analyzes the differences of simulation results between G21 and G00, and evaluates the possible changes brought to the GRAPES_GFS. But an in-depth analysis how the simulation results can improve the forecast performance is absent in this paper. The reason is that the forecast biases of the numerical model result from a combination of various factors, and it is



difficult to explain the improvement of the GRAPES_GFS forecast system just with a single case. Therefore, a batch of experiments are needed later in our future study. Since the absence of cloud-field information at a single time can bring systematic biases to the simulated temperature field and geopotential height field, in the cycling numerical forecasting operational system, the cloud-field information that has formed should be retained as much as possible. Moreover, the temperature and humidity structure in the initial field, especially the water vapor, can significantly affect the dynamic and

physical processes in the numerical model. Thus, in addition to the improvement of dynamic and physical processes, more attention should be paid to the assimilation of water vapor data, to improve the data quality of water vapor in the initial field of GRAPES_GFS.

*Code and data availability*. The model simulation data used in this study is available at https://pan.baidu.com/s/10lz_qEUwcKJXjinaFHy8BQ with access code 4w03; the model code is only available by request

via mazs@cma.gov.cn due to the confidential requirement.

*Author contribution*. ZSM and CFZ designed the experiments and ZSM carried them out. ZSM developed the model code and performed the simulations. ZSM prepared the manuscript with contributions from all co-authors.

**Acknowledgments**

This study was supported by the National Key R&D Program on Monitoring, Early Warning and Prevention of Major

Natural Disasters (grant 2017YFC1501406 and 2017YFC1501403), the National Natural Science Foundation of China (grants 41925022, 91837204, 41575143), the State Key Laboratory of Earth Surface Processes and Resource Ecology, and the Fundamental Research Funds for the Central Universities. We thank Nanjing Hurricane Translation for reviewing the English language quality of this paper.

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



**Figures**

Figure 1. Time evolution of global mean of the total water vapor tendency (WVT) and total temperature tendency (TT) at different vertical levels from 0 to 12h simulated by F00 (a,b), G21(c,d) and G00(e,f) experiments. The unit of WVT and TT is g/kg/day and K/day, respectively.

Figure 2. Time evolution of mean water vapor tendency (WVT) of the dynamical core and each physical processes at 300hPa, 500hPa and 925hPa heights from 0 to 1h simulated by F00 (a,d,g), G21(b,e,h) and G00(c,f,i) experiments. Unit: g/kg/day.

Figure 3. Same as Fig. 2, but for the results of temperature tendency. Unit: K/day.

Figure 4. Vertical distribution of total number of cloud points at different forecast time simulated by F00 (a), G21(b) and G00(c) experiment, respectively. Unit: number*10000.

Figure 5. Distributions of all hydrometeor content at 400hPa at different forecast time (5min, 1h, 3h, 6h) simulated by F00 (a-d), G21(e-h) and G00(i-l) experiments, respectively. Unit: g/kg.

Figure 6. Distribution of water vaper content (WVC) (a-b) simulated by F00 and G21, and the differences of WVC (c) and relative humidity (RH) (d) between F00 and G21 (F00-G21) at 400hPa in their initial fields. The units of WVC and RH are g/kg and %, respectively.

Figure 7. Zonal means and their differences of (a) 3h-averaged cloud water path (CWP), and (b) the outgoing longwave (OLR) at the top of atmosphere and the downward longwave at ground (GDLW) simulated by G21 and G00 experiments for 590 00~03UTC, August 9, 2019. The units of CWP and OLR/GDLW are g/m$^2$ and W/m$^2$, respectively.

Figure 8. Distribution of the differences (G00 minus G21) of temperature field (a-d) and geopotential height field (e-h) at 500hPa simulated by G00 and G21 experiments. The units of temperature and geopotential height are K and gpm, respectively.

Figure 9. Time evolution of the sum of averaged cloud water path and ice water path at the typhoon "LEKIMA" region 595 (117°E-130°E,22°N-34°N) simulated by G00 and G21 experiment. Unit: g/m$^2$.

Figure 10. Distribution of the differences (G00 minus G21) of 3-hourly and 24h accumulated precipitation (since 00UTC August 8, 2019) of the typhoon "LEKIMA" simulated by G00 and G21 experiments. Unit: mm.

Figure 11. Time evolution of the forecasted track errors of G00 and G21 experiments for the typhoon "LEKIMA" and "KROSA" during the forecast period of 72h. Unit: km.

**Tables**

Table 1. Model setup of three experiments used in this study.



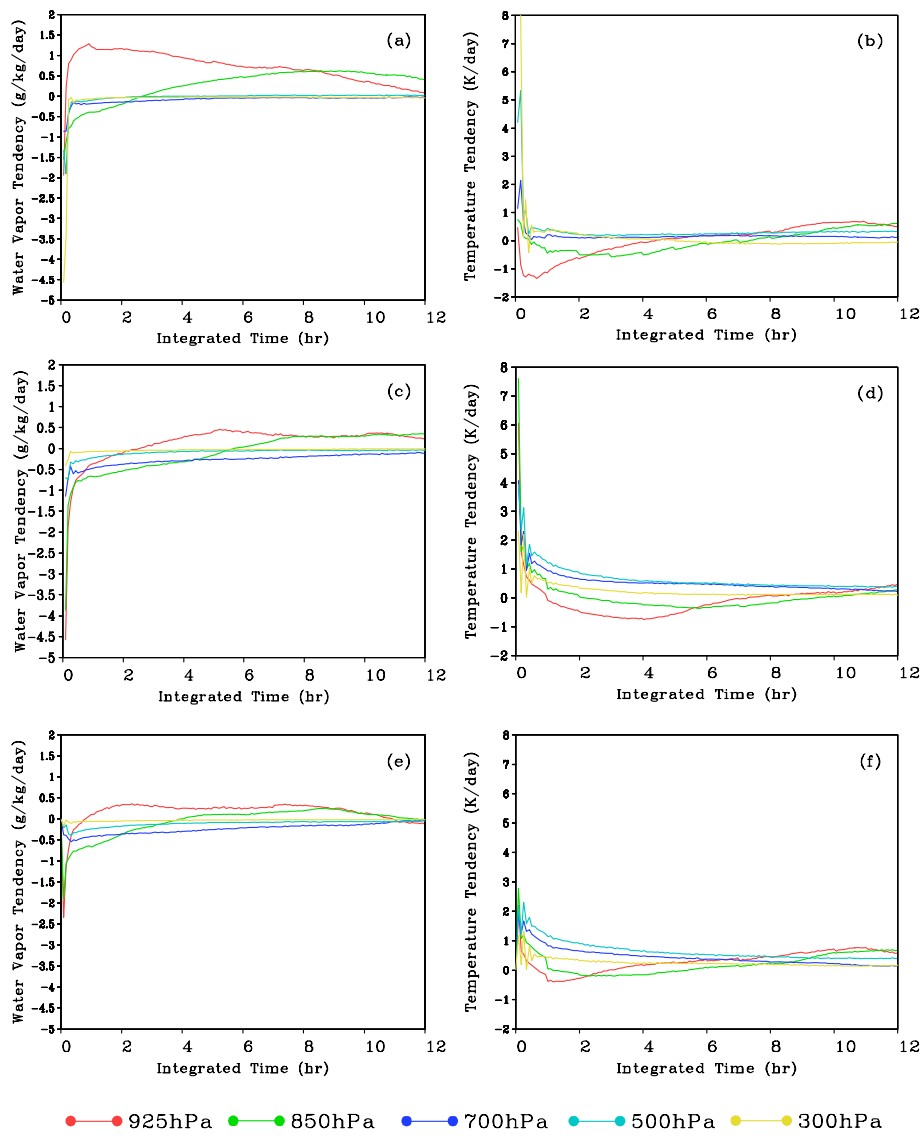

**Figure 1. Time evolution of global mean of the total water vapor tendency (WVT) and total temperature tendency (TT) at different vertical levels from 0 to 12h simulated by F00 (a,b), G21(c,d) and G00(e,f) experiments. The unit of WVT and TT is g/kg/day and K/day, respectively.**





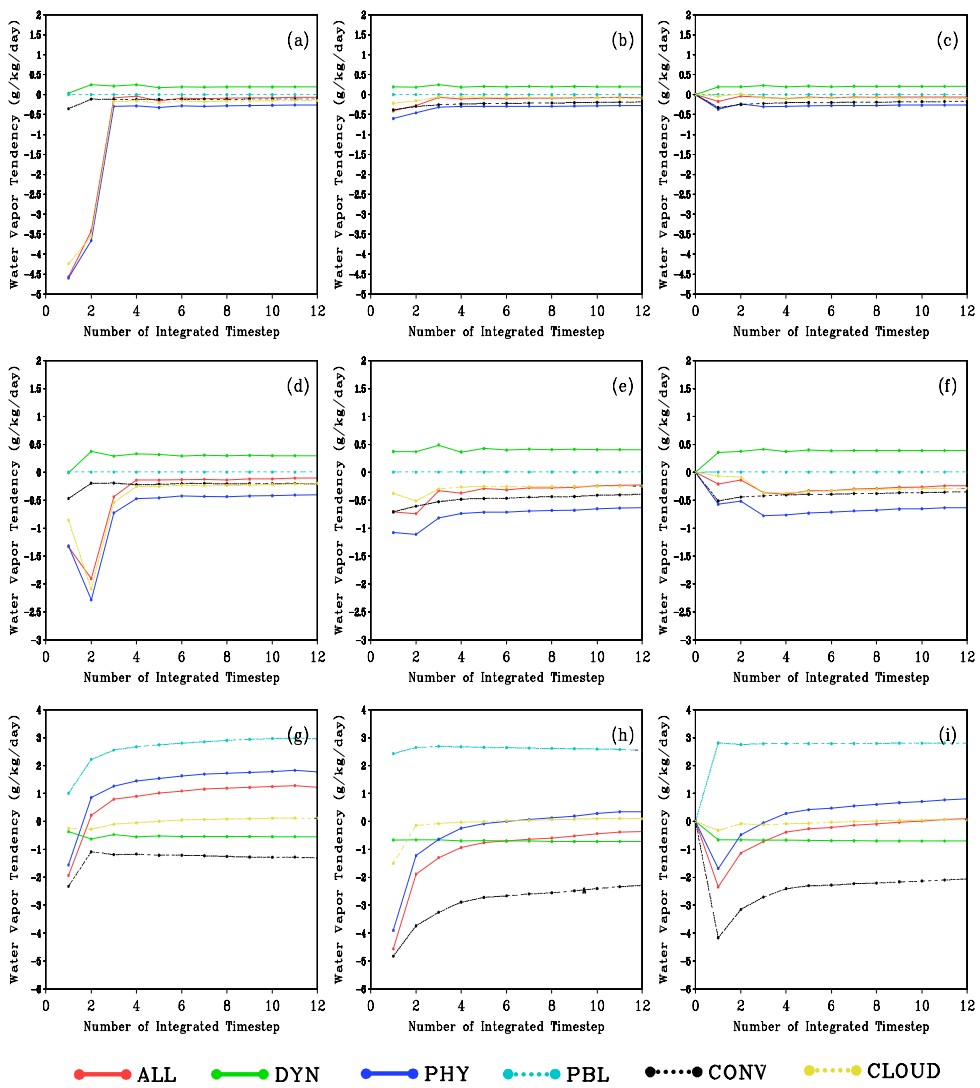

**Figure 2. Time evolution of mean water vapor tendency (WVT) of the dynamical core and each physical processes at 300hPa, 500hPa and 925hPa heights from 0 to 1h simulated by F00 (a,d,g), G21(b,e,h) and G00(c,f,i) experiments. Unit: g/kg/day.**




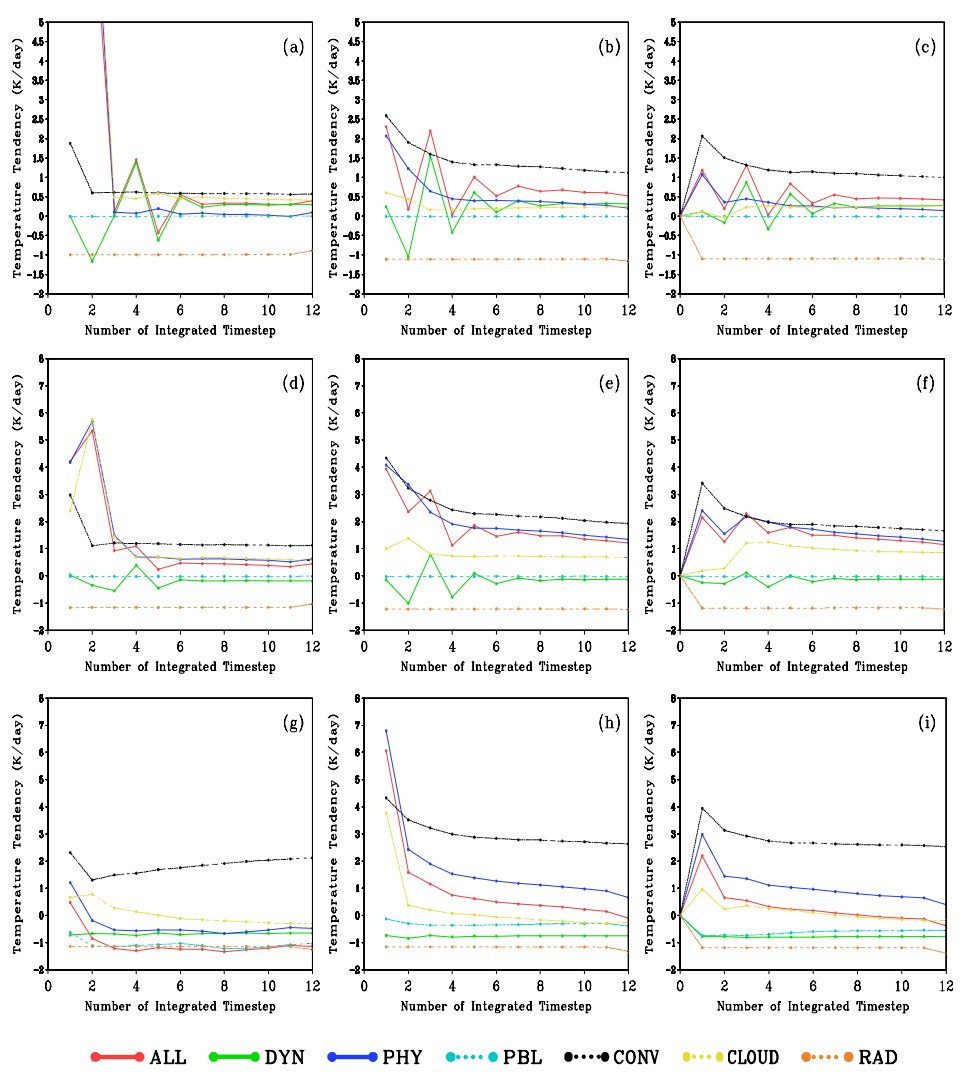

**Figure 3. Same as Fig. 2, but for the results of temperature tendency. Unit: K/day.**




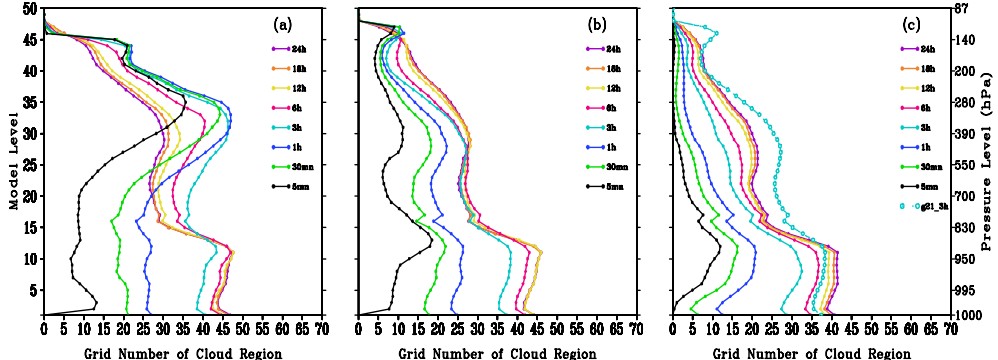

**Figure 4. Vertical distribution of total number of cloud points at different forecast time simulated by F00 (a), G21(b) and G00(c) experiment, respectively. Unit: number*10000.**




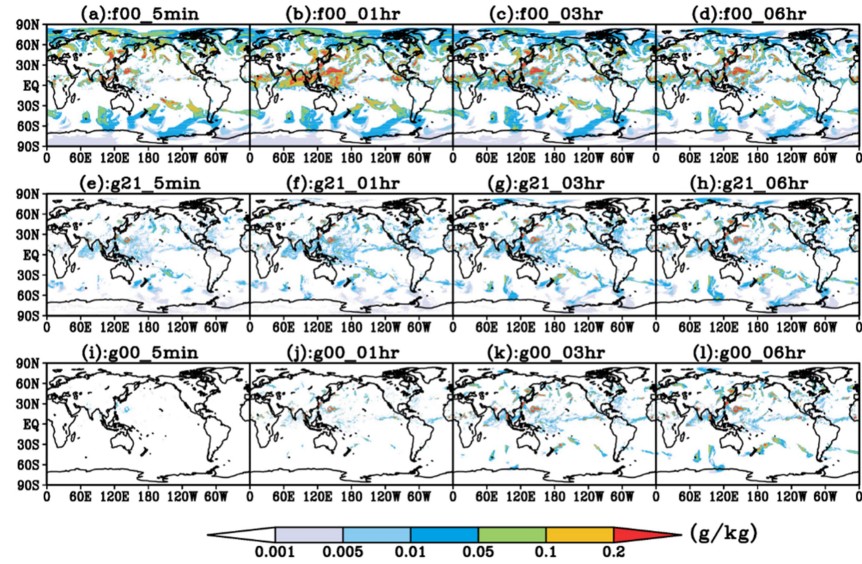

**Figure 5. Distributions of all hydrometeor content at 400hPa at different forecast time (5min, 1h, 3h, 6h) simulated by F00 (a-d), G21(e-h) and G00(i-l) experiments, respectively. Unit: g/kg.**


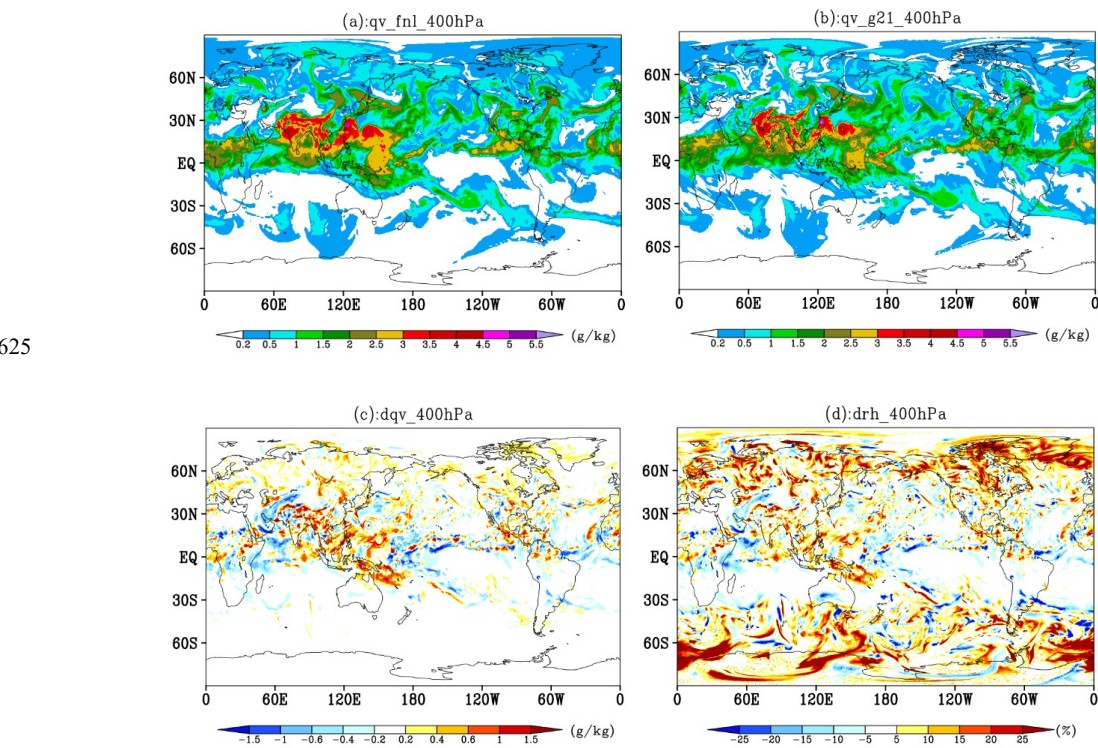

**Figure 6. Distribution of water vaper content (WVC) (a-b) simulated by F00 and G21, and the differences of WVC (c) and relative humidity (RH) (d) between F00 and G21 (F00-G21) at 400hPa in their initial fields. The units of WVC and RH are g/kg and %, respectively.**




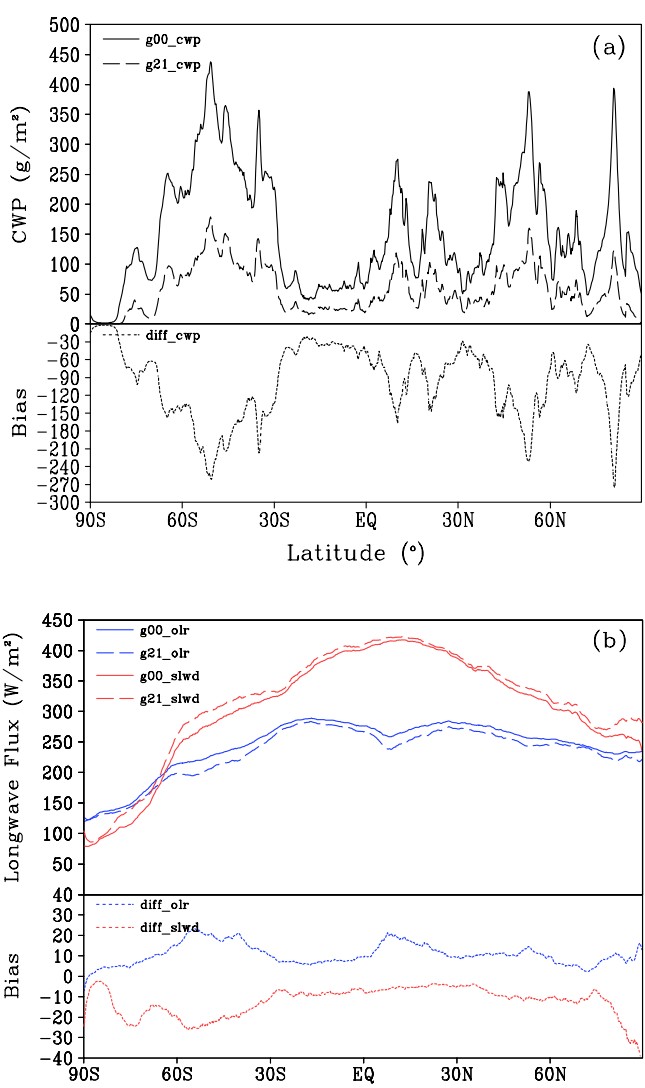

**Figure 7. Zonal means and their differences of (a) 3h-averaged cloud water path (CWP), and (b) the outgoing longwave (OLR) at the top of atmosphere and the downward longwave at ground (GDLW) simulated by G21 and G00 experiments for 00~03UTC, August 9, 2019. The units of CWP and OLR/GDLW are g/m² and W/m², respectively.**





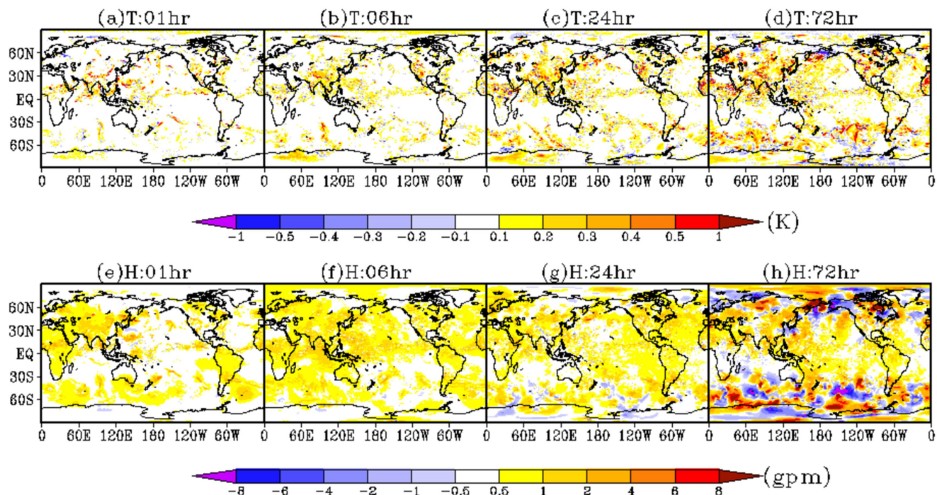

**Figure 8. Distribution of the differences (G00 minus G21) of temperature field (a-d) and geopotential height field (e-h) at 500hPa simulated by G00 and G21 experiments. The units of temperature and geopotential height are K and gpm, respectively.**



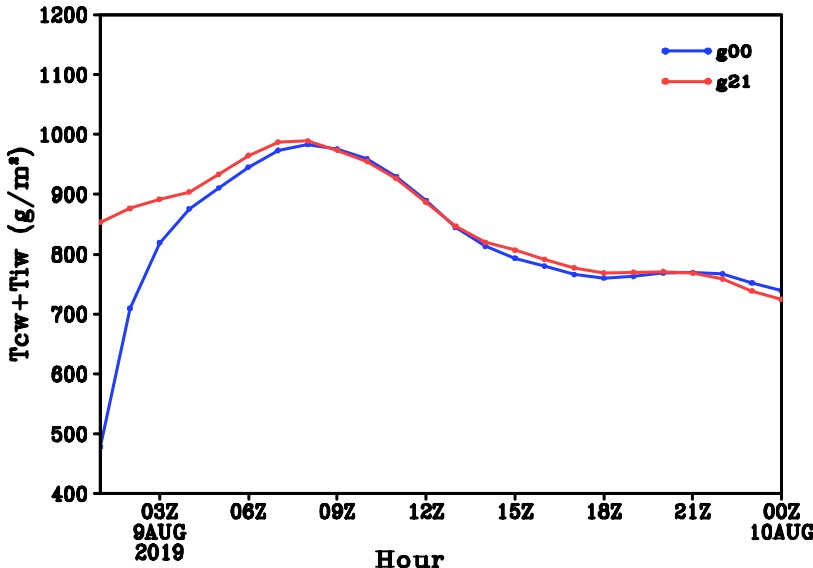


**Figure 9. Time evolution of the sum of averaged cloud water path and ice water path at the typhoon "LEKIMA" region (117°E-130°E,22°N-34°N) simulated by G00 and G21 experiment. Unit: g/m².**

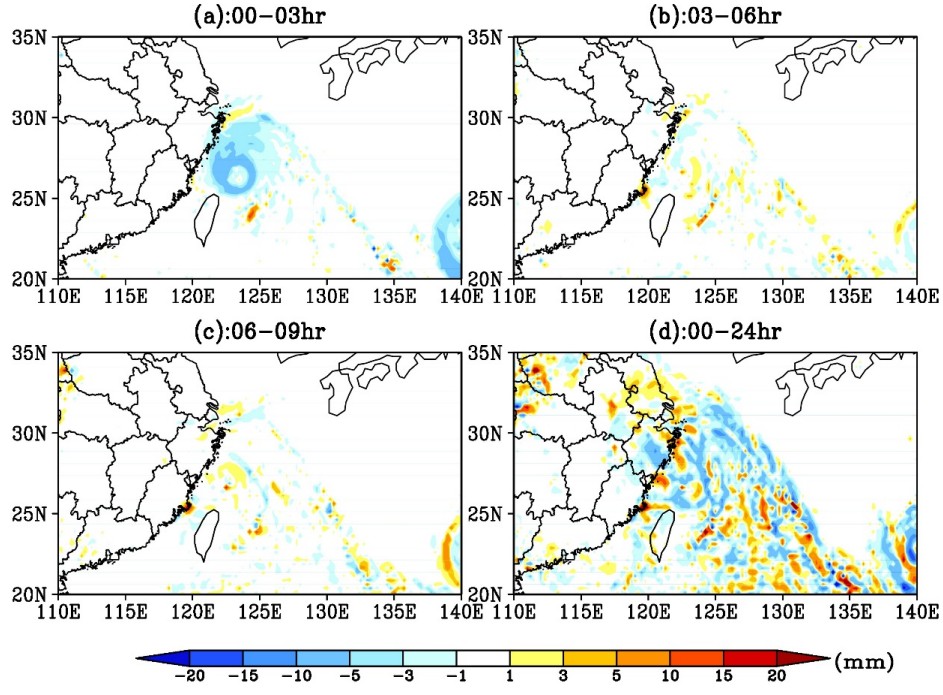


**Figure10. Distribution of the differences (G00 minus G21) of 3-hourly and 24h accumulated precipitation (since 00UTC August 8, 2019) of the typhoon "LEKIMA" simulated by G00 and G21 experiments. Unit: mm.**



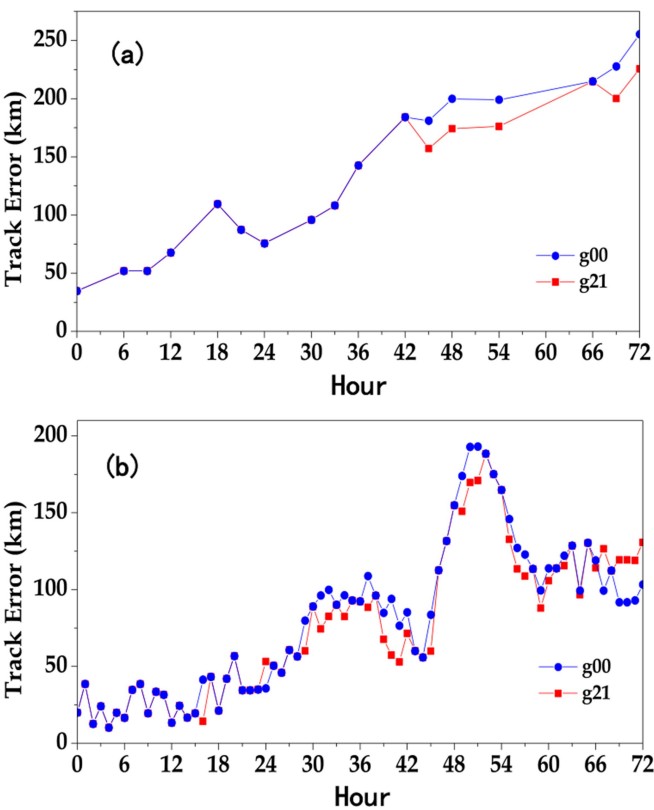

Figure 11. Time evolution of the forecasted track errors of G00 and G21 experiments for the typhoon "LEKIMA" and "KROSA" during the forecast period of 72h. Unit: km.





**Table 1. Model setup of three experiments used in this study**

| Experiment Name | Initial Field | Initial Forecast Time | Lead Time (h) |
|---|---|---|---|
| **G21** | 4D-VAR analysis fields | 2100 UTC, August 8, 2019 | 75 |
| **G00** | 4D-VAR analysis fields plus 3-hour integration | 0000 UTC, August 9, 2019 | 72 |
| **F00** | FNL reanalysis data | 0000 UTC, August 9, 2019 | 72 |