# Peer review of "Spin-up Characteristics with Three Types of Initial Fields and the Restart Effects on the Forecast Accuracy in GRAPES Global Forecast System"

_Geoscientific Model Development, 2020_

## Referee Comment (RC1) · Anonymous Referee #1 · 17 Jul 2020

Overview:

This paper examines the impact of initial conditions on the spin-up process in the GRAPES_GFS model, with results showing that the external FNL analysis is inferior to the model's internal analysis and that the removal of hydrometeor information during the cycling process has a deleterious effect and necessitates further spin-up time. These conclusions are convincingly examined from numerous angles, although the findings are somewhat as expected. Because of that, the paper would benefit from a bit more explanation of the motivation of the work (e.g., did the FNL used to be used prior to the 4DVAR upgrade? Why is it that the hydrometeor data is wiped out during the

cycling process, and is changing that currently under consideration?). There are also a few spots in the analysis, particularly of the tendencies, where there appear to be a few errors in labeling/incongruencies with the cited figures or sections that are unclear. A few of the figures could also stand to be a bit clearer in their labeling and font size. Finally, while the paper's writing is fairly good, there remain widespread instances of misused articles and awkward words and phrases that sometimes obscure the clarity of what is trying to be conveyed, some of which are noted in the technical corrections below. That said, the motivation of the work is sound and the analysis appears to be solid, so pending the specific comments listed below I believe the manuscript should be published.

Specific Comments:

Line 25: Changing "variation amplitudes" to "variations in amplitude…" may be clearer here.

Lines 81, 144: By "resolution", do the authors actually mean the "grid spacing"?

Lines 95-102: If I am understanding correctly, the operational model includes 3 hours of "built in" spin-up time so that forecasters looking at a launched forecast don't have to discard the first few hours of the model run. If that is the case, why is it that the hydrometeor variables are discarded? Is it due to limitations of disk space during the restart? Because this seems like something that would predictably introduce problems and negate the benefits of spinning the model up earlier (which the results of the paper confirm), I think it would be helpful for the authors to provide a bit of history about why this is currently done.

Line 129: Consider removing "weather process" here, as I think the sentence reads more clearly without it.

Lines 134-135: The sentence beginning "In the second experiment" is quite unclear to me. What "results" are being talked about here? How is an "initial field" used?

The following sentence is clearer in terms of what is actually being done, so consider rephrasing or removing.

Line 187: What is meant by the "co-action of cloud and convections" here? Is this co-action shown in the figure?

Lines 195-196 and elsewhere: Does "physical process" here refer to the line labeled PHY in Fig. 2, or all 'physical processes' in the model (versus dynamic)? Make sure this is clear throughout the text. The text also states that the biggest difference between F00 and G21 is caused by the convection scheme, but it appears to me that the PHY line is also significantly different between the two.

Section 3.1.2 overall: Related to the previous comment, it would be helpful if the authors state more explicitly how each of the tendencies in Figs 2, 3, etc are defined. 'RAD' and 'PBL' are somewhat straightforward, but the authors should state clearly where the CLOUD, CONV, and PHY tendencies are coming from and how they all differ.

Line 211: Please add reference to Figure 3 here.

Line 212: Again, related to comments 7 and 8, the authors state that it is due to "Cloud and convection processes", but Fig. 2a,d seems to show the biggest changes due to CLOUD and PHY rather than CONV (black line). Please clarify.

Line 222: It appears to me that the DYN and particularly the PHY line in Fig. 3g show much smaller adjustments than the middle and upper levels, not larger. Please clarify.

Line 223: Re: "dehumidifying and heating of the atmosphere", doesn't Fig. 3g show an overall cooling of the atmosphere (negative TT for the ALL line), corresponding to a generally positive overall WVT in 2g?

Line 237: It appears to me that the adjustments in G00 are almost half those in G21 at all levels (at least for the first few integrations) and not what I would characterize as "close to or slightly smaller".

Line 248: Please define how the total hydrometeor content is defined (even if it is just cloud + ice + rain + snow + hail, etc.).

Lines 258-259: Is that how the equilibrium state is being defined overall, or is the description given here (24 hours vs. 6 hours below 850 hPa) only for this case? It seems more accurate to state that the equilibrium state is defined as when the difference with respect to the 24-hour integration is minimal, implicitly assuming that equilibrium will have been reached by 24 hours. Also, please quantify what "is insignificant" means here. Is it just being used subjectively?

Lines 251-267: Can the authors add some discussion of the equilibrium "overshoot" in F00 at upper levels? This was one of the more noteworthy things I noticed about this figure.

Line 292: Should "initial time" really be 21Z (i.e., in G21)?

Lines 308-309: These findings are definitely in agreement with past studies about the importance of an accurate initial moisture field, at least on the storm-scale. It may benefit the paper and further emphasize the authors' point to add some references to other papers discussing the importance of accurate moisture DA, e.g.:

Weygandt, S. S., A. Shapiro, and K. K. Droegemeier, 2002: Retrieval of model initial fields from single-Doppler observations of a supercell thunderstorm. Part II: Thermodynamic retrieval and nu- merical prediction

Ge, G., J. Gao, and M. Xue, 2013: Impacts of assimilating measurements of different state variables with a simulated supercell storm and three-dimensional variational method.

Line 329: Do the authors mean the G21 run instead of observations? If not, what observations are being referenced here?

Line 352: This sentence is unclear to me as I don't understand what is meant by "same forecasts", although I assume the authors are stating that the conclusions for
both Lekima and Krosa are the same and therefore only Lekima will be presented. Please clarify.

Line 355: Can "CCWV" and "TCIW" be made consistent with the axis labels in Fig. 9, of vice versa?

Figures 1, 2, 3: Tick labels are small and hard to read. Please enlarge.

Figure 4: Legend text is too small to read.

Figures 1, 2, 3, 4: Please add titles to each subplot of what run, height, time, etc. are being shown in each panel. It is confusing having Figure 1 vary by run in each row, Figure 2 vary by run in each column, etc.

Figure 7a: Are the legend labels switched here? As per the discussion, shouldn't g21_cwp be higher than g00_cwp?

Technical Corrections:

Line 32, 37, elsewhere: Change "Besides" to "In addition"

Line 43: Change "reasonability" to "representativeness"

Line 53: "model" → "modeling"

Line 56: "could" → "can"

Line 91: "widely-used" → "widely used"

Line 115: "difference" → "differencing"

Line 128: Should "1.2" be "2.2" here?

Line 143: "outputted" → "output"

Line 145: "operational solution" → "the operational setup"

Line 182: "in the" → "due to"

[Figure]

Lines 185-186: "this level" → "these levels"

Line 187, 198, 202, 208, elsewhere: "convections" → "convection"

Line 244: Remove "analysis"

Line 251: "no matter the" → "regardless of whether the"

Line 273: "lead time" → "forecast time"

Line 277: "at time 1 hour after" → "at 1 hour into"

Line 302: "can reflect" → "reflects"

Line 303: Remove "relatively"

Line 305: "It" → "This"

Line 312: "in the operation" → "operationally"

Line 313: "less THC" → "decreased THC"

Line314: Remove "situation"

Line 315, 351: "typhoon track" → "track"; "landed on" → "made landfall in"

Line 327: "cloud" → "clouds"

Line 334: "moments" → "times"

Line 335: "two" → "four"

Line 351: "continued to develop on ocean" → "remained offshore"; remove "from off-shore areas"

Line 360: "strengthening" → "increase"

Line 368: "get" → "gets"; remove "of them"

Line 371: "an alternation of" → "alternating"

Line 374, 381: "While" → "In contrast,"

Line 382: Should "G20" here by "G21"?

Line 388: Should "Lichma" by Lekima?

Line 398: "All the three different experiments" → "All three experiments"

Line 407: "unobvious" → "not obvious"

Line 418: "analysis" → "analysis of"

―――――――――――――――――

---

## Short Comment (SC1) · 28 Aug 2020

This is an executive editor comment highlighting the ways in which this manuscript is not currently compliant with GMD policy on code and data availability.

The issues here must be addressed before a revised manuscript can be accepted for publication:

1. No model source code. The code and data availability section merely states that code is not available due to "the confidentiality requirement". GMD does permit model code to be withheld from publication if this is unavoidable for reasons be-

yond the control of the authors. Usually this is because the copyright licence of the code does not permit redistribution. However, the reasons that the authors cannot release the code must be detailed in the code and data availability section. In particular, it is important to state who can get a licence and how.

2. Version not identified. Neither the title of the manuscript nor the code and data availability section state precisely which version of GRAPES_GFS was used. This makes it impossible to reproduce the work even if one has a licence.

3. Model data is not on a persistent public repository. The model data appears to be on a cloud storage provided by Baidu. This lacks the persistence, non-revocability and persistent identifiers required for a journal publication. The data should instead be stored in a properly persistent archive with a persistent identifier such as a DOI. I note in this regard that the authors are from national laboratories so I would expect such facilities to be available to them.

4. No configuration, run, or data processing scripts. The configuration files, run scripts and any data processing or analysis scripts used to produce the results presented in the manuscript need to be publicly and persistently archived, and cited from the code and data availability section. As a guide, every file the user would need to reproduce the manuscript should accessible.

Further details on code and data availability requirements are in the GMD model code and data policy: https://www.geoscientific-model-development.net/about/code_ and_data_policy.html. The reasons for the policy and more detail are provided in this editorial: https://doi.org/10.5194/gmd-12-2215-2019. In particular, these documents cover what information you need to provide if the model source code can not be released, and what the requirements of suitable data archives are.

---

## Referee Comment (RC2) · Anonymous Referee #2 · 12 Oct 2020

The manuscript investigates the influence of spin-up and restart in a global weather forecast system GRAPES. Such a topic is important, as careful handling of those technical issues can greatly improve the accuracy of weather prediction. By comparing different spin-up and restart methods, the authors gain important knowledge of the forecasting system, such as that the GRAPES with its own analysis field performs better than the one using NCAR final reanalysis (FNL) data for the cold start in the spin-up. The paper contains useful information for model development and usage. I recommend its publication with GMD, pending on some minor comments below.

About the experiment setup. To better illustrate the differences between three experi-

[Figure]

ments, which are of great importance to this paper, can the authors use a schematic plot to show how the three runs were performed and what are the key input data. Also, it should be explicitly stated in the Section 1.2, why those three experiments were conducted, or in other words, what we expect to learn by comparing them. The three-hour lag confuses me a little bit.

It is unclear to me what is the current protocol for spin-up and restart strategies used by CMA that uses GRAPES_GFS to conduct the daily weather prediction as well as the extreme weather prediction like typhoons. According to this study, is there any modification needed on the protocol?

Is the total grid number of cloud (TGNC) related with the total cloud fraction? The latter is a more common term. Also, 1.0 e-4 g kg$-1$ threshold of cloud sounds an arbitrary choice. Are the results sensitive to this threshold definition?

Fig. 11a, why g00 and g21 are identical before 42 Hour and then become different abruptly?

---

## Author Comment (AC1) · 5 Nov 2020

Response to Reviewer 1:

We thank the anonymous referee for his/her valuable comments and suggestions that have helped us improve the paper quality. Our detailed responses (**Bold**) to the reviewer's questions and comments (*Italic*) are listed below.

**Anonymous Referee #1:**

*Overview:*
*This paper examines the impact of initial conditions on the spin-up process in the GRAPES_GFS model, with results showing that the external FNL analysis is inferior to the model's internal analysis and that the removal of hydrometeor information during the cycling process has a deleterious effect and necessitates further spin-up time. These conclusions are convincingly examined from numerous angles, although the findings are somewhat as expected. Because of that, the paper would benefit from a bit more explanation of the motivation of the work (e.g., did the FNL used to be used prior to the 4DVAR upgrade? Why is it that the hydrometeor data is wiped out during the cycling process, and is changing that currently under consideration?). There are also a few spots in the analysis, particularly of the tendencies, where there appear to be a few errors in labeling/incongruencies with the cited figures or sections that are unclear. A few of the figures could also stand to be a bit clearer in their labeling and font size. Finally, while the paper's writing is fairly good, there remain widespread instances of misused articles and awkward words and phrases that sometimes obscure the clarity of what is trying to be conveyed, some of which are noted in the technical corrections below. That said, the motivation of the work is sound and the analysis appears to be solid, so pending the specific comments listed below I believe the manuscript should be published.*

**We highly appreciate the reviewer's positive evaluation about this study. We also thank the reviewer for the valuable and detailed comments and suggestions which have helped us improve the paper quality.**

**For the suggestion in the overview, we added/revised two more detailed explanations in corresponding section (Line 89 and Lines 95 - 102, respectively) to clarify the motivation of this study.**
**(1) For the suggestion that "(e.g., did the FNL used to be used prior to the 4DVAR upgrade?)".**
**No, FNL data was not used as initial field in the GRAPES_GFS operation. However, it used to be used in the model research and development. For the GRAPES-GFS batch experiment, the cold-start simulation tends to consume less computing resources than cycle assimilation simulation, and the developers can faster obtain their wanted results. Therefore, in the development/modification of GRAPES_GFS, FNL data as the model's initial field is usually first adopted to simulate and evaluate the impact of modification in the dynamic core and physical process on forecast performance. When the result is ideal, then the cycle experiment with 3D-VAR/4D-VAR is carried out and its forecast**

**output is used to analyze the final effect of the modification.**

**We added the following sentence in Line 91 to describe how the FNL data is used in research and development of the GRAPES-GFS.**
**"In the research and development of the GRAPES-GFS, the widely-used FNL (Final Operational Global Analysis) reanalysis data provided by NCEP (National Centers for Environmental Prediction) (Kalnay et al., 1996) is usually adopted as the model's initial field to quickly evaluate the effects of modification in dynamic core and physical processes on the model forecast performance, because the cold start simulation with FNL consumes less computing resources than that of cycle assimilation simulation. What advantages does the new 4D-VAR assimilation analysis fields have in spin-up process compared with the cold start simulation with FNL?"**

**(2) For the question "Why is it that the hydrometeor data is wiped out during the cycling process, and is changing that currently under consideration?"**
**we gave a detailed reply (as reply to comment 3) and revised the corresponding section in Lines 95 - 102.**

**Specific Comments:**

*Line 25: Changing "variation amplitudes" to "variations in amplitude…" may be clearer here.*
**We agree with the reviewer and changed Line 25 to "…, because the variations in amplitude of the temperature and humidity tendency..."**

*Lines 81, 144: By "resolution", do the authors actually mean the "grid spacing"?*
**Yes, the "resolution" in Lines 81 and 144 means "grid spacing", which is more specific and accurate. We changed "resolution" to "grid spacing" in the corresponding lines.**

*Lines 95-102: If I am understanding correctly, the operational model includes 3 hours of "built in" spin-up time so that forecasters looking at a launched forecast don't have to discard the first few hours of the model run. If that is the case, why is it that the hydrometeor variables are discarded? Is it due to limitations of disk space during the restart? Because this seems like something that would predictably introduce problems and negate the benefits of spinning the model up earlier (which the results of the paper confirm), I think it would be helpful for the authors to provide a bit of history about why this is currently done.*
**It is a good suggestion. We agree with the reviewer and it would be helpful to improve the understanding for the motivation/background of the study when we give a further introduction why the cloud-field variables are discarded and provide the reasons why this is currently done.**
**For users (especially forecasters) of numerical weather prediction models, they are usually accustomed to using the forecast productions of numerical models staring to integrate from 00 UTC or 12 UTC (or more time, 06 UTC, 18 UTC). We adjusted the operational process of GRAPES_GFS to adapt to user's usage habit. We take 00 UTC as**

an example for explanation. GRAPES_GFS uses the 4D-Var assimilation system to improve the initial field quality. For 00 UTC, 4D-var assimilation system needs to perform data assimilation analysis using the model meteorological fields of the first three hours (21 UTC), and finally generate assimilation analysis field at 21 UTC, that is, the initial field of the G21 experiment. Considering the user's habits, the model at 00 UTC is terminated after three hours integration from 21 UTC. In the process, only the essential meteorological field variables (u, v, th, qv, pi, ps, etc.) are retained for restart, while the existing cloud-field variables (the mass and concentration of hydrometeors and cloud cover) are discarded. The model restarts using the above retained variables at 00 UTC, and its forecast products are used to release to users, which is the G00 experiment. The reasons for the unretained cloud field variables are mainly based on the following considerations: the hydrometeor contents are very small amount relative to water vapor and they can be quickly created in the spin-up process when the model restarts. Moreover, it can save storage space and IO time. However, there are no studies that have carefully analyzed and evaluated its impact on the spin-up process and model forecasts. This is the focus of this paper.

We revised the content in Lines 95 - 102 to more clearly describe the changes of model forecast variables during the terminated-restart process of GRAPES-GFS and the reasons why the impact of the loss of cloud-field variables can be accepted in operation. "Actually, for numerical weather prediction model's users (especially forecasters), they are usually accustomed to referring the forecast productions of numerical models staring to integrate from 00 UTC or 12 UTC (or more time, 06 UTC, 18 UTC). Thus, considering the habit of users in using the forecast results, GRAPES_GFS integrates for 3 hours (to 1200 UTC) to retain the essential meteorological element fields (U, V, T, Q, H, TS , Ps, etc.), and then the integration is terminated and restarts from 1200 UTC by using the new-saved meteorological field data. The model forecast results thereafter are released, that is, the forecast results at 1200 UTC are obtained by users. In this process, the cloud-field variables (the mass and concentration of hydrometeors and cloud cover) during the first 3 hours of integration are not retained in the model, losing the cloud information formed after the 3-hour spin-up. The reasons for the unretained cloud-field variables were mainly based on the following considerations: the hydrometeor contents are very small amount relative to water vapor and they can be quickly created in the spin-up process when the model restarts. Moreover, this treatment can save storage space and input/output (IO) time. However, its impacts on the spin-up process and the model forecast performance have not yet been carefully analyzed and evaluated."

*Line 129: Consider removing "weather process" here, as I think the sentence reads more clearly without it.*
We agree with the reviewer and removed "weather process in" in Line 129.

*Lines 134-135: The sentence beginning "In the second experiment" is quite unclear to me. What "results" are being talked about here? How is an "initial field" used? The following sentence is clearer in terms of what is actually being done, so consider rephrasing or*

*removing.*

We agree with the reviewer that the description for the second experiment is not too clear. Here, the "results" means the forecast output from the second experiment, and we originally wanted to express: the model output of the second experiment is exactly the forecast results to be provided to users by GRAPES_GFS in operation. The "initial field" of the second experiment has been explained in the rephrased section. According to the reviewer's suggestion, we rephrased the description for the second experiment as follows:

"For the second experiment, called G00, its initial field adopts 3 hours integration output of G21 without retaining cloud-field information. That is to say, at 0000 UTC on August 9, it retains the G21's 3-h forecast variables (u and v wind field components, potential temperature, water vapor and dimensionless air pressure, etc.) required by the pre-processing system and stops the integration. During the process, the fields of all hydrometeor contents and cloud cover are lost considering the limitation of IO time and disk space. And then the model restarts at 0000 UTC on August 9 with the reserved forecast-field information for forecasting in G00. Moreover, the model output of G00 is exactly the forecast results to be provided to users in the GRAPES_GFS operation."

*Line 187: What is meant by the "co-action of cloud and convections" here? Is this co-action shown in the figure?*

It is a good suggestion, and we provide further explanation in the section. In the GRAPES_GFS, the variation of water vapor is determined by dynamic core (DYN), turbulent mixing of planetary boundary layer (PBL), cumulus convection process (CONV) and cloud physical process (CLOUD). Among them, the sum of the last three items is referred to as the total tendency of all physical processes (PHY). In Figs. 2b and 2e, we can see that the variations of WVTs from DYN and PBL are much smaller compared with CONV and CLOUD, thus we originally described it in this section "…, while the variation of water vapor is mainly caused by the co-action of cloud and convections". To describe this part more clearly, we rephrased it as follows.

"At these levels in G21 (Figs. 2b and 2e), the total WVTs at the first few integration steps are slightly larger than that at the subsequent integration steps. The variations of the WVTs from dynamic core and turbulent mixing process in the planetary boundary layer are much less than those from the cumulus convection process and cloud physical process, and the latter two processes jointly determined the variation of WVTs at 300 hPa and 500 hPa."

*Lines 195-196 and elsewhere: Does "physical process" here refer to the line labeled PHY in Fig. 2, or all 'physical processes' in the model (versus dynamic)? Make sure this is clear throughout the text. The text also states that the biggest difference between F00 and G21 is caused by the convection scheme, but it appears to me that the PHY line is also significantly different between the two.*

We agree with the reviewer's comment and it is a good suggestion. Yes, the "physical process" throughout the text refers to "all physical processes" (versus dynamic core) in the GRAPES_GFS. We checked throughout the text and changed the "physical process"

to "all physical processes" in order to clarify its exact meaning, which means the total tendency of all physical processes in the model.

Since PHY represents the total water vapor tendency of all physical processes, it includes the contribution of the convection scheme, thus the PHY lines between F00 and G21 also takes roles on significant difference. Their difference of WVT between the two experiments is essentially caused by convection scheme (Figure 2g and 2h).

In addition, at the beginning of section 3.1.2, we added the following description to state the compositions of tendency of temperature and water vapor in the GRAPES_GFS, which can benefit to a better understanding.

"In the GRAPES_GFS, the total temperature tendency of the model (ALL) is determined by dynamic core (DYN), radiation process (RAD), turbulent mixing in planetary boundary layer process (PBL), cumulus convection process (CONV) and cloud physical process (CLOUD). Among them, the total temperature tendency of all physical processes (PHY) is defined as the sum of the last four items (PHY=RAD+PBL+CONV+CLOUD). Likewise, the total water vapor tendency for ALL and PHY are same to those of temperature tendency except for the radiation process (RAD). "

*Section 3.1.2 overall: Related to the previous comment, it would be helpful if the authors state more explicitly how each of the tendencies in Figs 2, 3, etc are defined. 'RAD' and 'PBL' are somewhat straightforward, but the authors should state clearly where the CLOUD, CONV, and PHY tendencies are coming from and how they all differ.*
We agree with the reviewer. Except for the reply to the previous comment, we added corresponding description for all abbreviations (ALL, PHY, RAD, PBL, CONV and CLOUD) in Figs 2, 3 to state them clearly. The modifications for Figs 2 and Figs 3 are as below:

"Figure 2. Time evolution of mean water vapor tendency (WVT) of dynamical core (DYN), planetary boundary layer process (PBL), cumulus convection process (CONV), cloud physical process (CLOUD), the total of all physical processes (PHY, PHY=PBL+CONV+CLOUD) and the total of the model (ALL, ALL=DYN+PHY) at 300hPa, 500hPa and 925hPa heights from 0 to 1h simulated by F00 (a,d,g), G21(b,e,h) and G00(c,f,i) experiments. Unit: g/kg/day."

"Figure 3. Time evolution of mean temperature tendency (TT) of dynamical core (DYN), radiation process (RAD), planetary boundary layer process (PBL), cumulus convection process (CONV), cloud physical process (CLOUD), the total of all physical processes (PHY, PHY=RAD+PBL+CONV+CLOUD) and the total of the model (ALL, ALL=DYN+PHY) at 300hPa, 500hPa and 925hPa heights from 0 to 1h simulated by F00 (a,d,g), G21(b,e,h) and G00(c,f,i) experiments. Unit: K/day."

*Line 211: Please add reference to Figure 3 here.*

**We added the figure reference in line 211.**

**"In the middle and upper layers of the model, the dramatic change of the TT in F00 mainly occurs within the first half hour of the integration (Figs. 3a and 3d)."**

*Line 212: Again, related to comments 7 and 8, the authors state that it is due to "Cloud and convection processes", but Fig. 2a,d seems to show the biggest changes due to CLOUD and PHY rather than CONV (black line). Please clarify.*

**We agree with the reviewer. It can easily cause errors in understanding if we do not clarify the meaning of CLOUD, CONV and PHY in Figure 2 and Figure 3. We modified the descriptions for Figures 2, 3 in the reply to comments 7 and 8. Because the temperature tendency (TT) of PHY in Figure 3 is the total TT related with physical processes, PHY=RAD+PBL+CONV+CLOUD, it is reasonable that the TT of PHY shows changes consistent with that of CLOUD that is biggest variation in amplitude among the physical processes shown in Figure 2a,2d and Figure 3a,3d.**

**For the changes of water vapor tendency (WVTs, Figure 2a,2d) and temperature tendency (TTs, Figure 3a,3d) in the F00 experiment, although the directions of their change conform to the physical laws (condensation process leads to negative water vapor tendency and positive temperature tendency, vice versa), their variations in amplitude are quite different. The variation in amplitude of TTs appears to be more dramatic than that of WVTs. Among all the TTs shown in Figure 3a and 3d, the cloud physical process is the biggest one in the first time step, followed by convection process. The TT and WVT of convection process decrease rapidly after one time step. Based on the above description, the original expression in line 212 is not appropriate and we modified it as follows:**

**"Among all the TTs at the first integration step, the cloud physical process leads to the biggest one, followed by convection process, and they are related with the water vapor condensation process (Figs. 2a and 2d)."**

**At the same time, we deleted the following sentence "Compared with the convection process, the cloud physical process can cause greater temperature adjustments." to keep the meaning coherent in this section.**

*Line 222: It appears to me that the DYN and particularly the PHY line in Fig. 3g show much smaller adjustments than the middle and upper levels, not larger. Please clarify.*

**Sorry for our mistake and thanks for helping figure it out. We corrected "a relatively large and rapid adjustment" to "a relatively small and rapid adjustment" in Line 222.**

*Line 223: Re: "dehumidifying and heating of the atmosphere", doesn't Fig. 3g show an overall cooling of the atmosphere (negative TT for the ALL line), corresponding to a generally positive overall WVT in 2g?*

**In line 223, we had wanted to describe the temperature tendency (TT) of the convection scheme first since it is different from other physical processes and dynamic core. We rephrased this section as follows:**

**"The TT of the convection process at 925 hPa in F00 varies between 1.5 K d$^{-1}$ and 2 K d$^{-1}$, which is mainly caused by condensing and dehumidifying of the atmosphere (Fig. 2g). Except for the cloud physical process with a relatively small positive tendency in the first four time steps, the TTs of dynamic core and other physical processes are all negative. Overall, in F00 the total atmospheric temperature is reduced with an amplitude of about -1.2 K d$^{-1}$ in the first hour of the integration at 925 hPa."**

*Line 237: It appears to me that the adjustments in G00 are almost half those in G21 at all levels (at least for the first few integrations) and not what I would characterize as "close to or slightly smaller".*

**We believe that the reviewer's description for the difference of TT between G00 and G21 in Line 237 is more accurate and we modified the sentence as follows:**
**"In the first few time steps, G00 also has an adjustment process, with the adjustment amplitudes of TT close to half those in G21 at all levels."**

*Line 248: Please define how the total hydrometeor content is defined (even if it is just cloud + ice + rain + snow + hail, etc.).*

**We agree with the reviewer and added the definition for the total hydrometeor content in the sentence of Line 248, "The total hydrometeors content (THC, THC = cloud water + raindrop + cloud ice + snow + graupel) greater than…"**

*Lines 258-259: Is that how the equilibrium state is being defined overall, or is the description given here (24 hours vs. 6 hours below 850 hPa) only for this case? It seems more accurate to state that the equilibrium state is defined as when the difference with respect to the 24-hour integration is minimal, implicitly assuming that equilibrium will have been reached by 24 hours. Also, please quantify what "is insignificant" means here. Is it just being used subjectively?*

**It is a good comment on the description for the definition of equilibrium state. We wanted to give an introduction of the equilibrium state for levels below 850 hPa in lines 258-259. Certainly, it would be more conducive to the quantitative analysis of spin-up time if the definition of equilibrium state applicable to all levels is given. In fact, it is difficult to quantitatively determine the time to complete spin-up process in model because of the comprehensive adjustment and changes from the dynamic and the physical processes. We have drawn the distribution of total grid number of cloud (TGNC) of 48-h and 72-h, and we found their vertical distributions are very close to that of 24-h. Therefore, we adopt 24-h TGNC as the reference standard to analyze the equilibrium time, which is what you mentioned "implicitly assuming that equilibrium will have been reached by 24 hours". We agree that the definition of equilibrium state you gave is more accurate and can be applicable to all levels. In our original definition, the time (after 6 hours) should not be a specific time that is used to compare with the TGNC at 24 hours. Based on the above analysis, we modified the definition of equilibrium state as follows:**
**"Note that the statistical equilibrium state is defined when the difference of TGNC with respect to the 24-hour integration is insignificant (the difference is less than 20% of**

**TGNC at 24-hour).”**

**We rechecked our descriptions in the section and think it is reasonable to consider 20% as insignificant difference in TGNC, which is consistent with the description for the spin-up time at all levels, while it seems a little subjective.**

*Lines 251-267: Can the authors add some discussion of the equilibrium "overshoot" in F00 at upper levels? This was one of the more noteworthy things I noticed about this figure.*

**Yes, we can and it is a good suggestion. As shown in the Fig.4a, the total grid number of cloud (TGNC) of the F00 at upper levels has a larger value than those of G00 and G21, which is mainly caused by the differences of humidity field in their initial fields. We take F00 and G21 as examples and added two additional figures (shown below) to state it. Compared with G21, F00 has a wetter water vapor environment at the upper levels (Fig.6d), which tends the water vapor to quickly condense into more hydrometeors at the beginning of the integration to eliminate supersaturated water vapor. When the model began to integrate the F00 experiment has a higher hydrometeor content value and a wider distribution of cloud region (Fig. R1 and Figs.5a and 5e), thus its TGNCs are also larger than those of G21 (Fig.4a) at the upper layers. For the cloud formation at the beginning of integration (0 to 3 timesteps), it is mainly completed through the condensation process in the cloud scheme, yet convection scheme has less contribution because it need take a certain time to reach the triggering condition and detrainment of the hydrometeors (Figs. R2a and R2b, Figs. R2a and R2d). With the integration of the model and the potential impact of advection process of dynamics on environmental humidity field, the clouds scheme is no longer dominated by the condensation process, but presents the coexistence of condensation and evaporation (Fig. R2c).**

**Based on the above analysis, we added the following discussion in Line 267:"Compared with G21, F00 has a wetter water vapor environment at the upper levels (Fig.6d), which tends the water vapor to quickly condense into more hydrometeors through cloud scheme to eliminate supersaturated water vapor at the beginning of the integration (Fig.2a). Thus F00 has a higher hydrometeor content value and a wider distribution of cloud region (Figs.5a and 5e) and its TGNCs are also larger than those of G21 at the upper layers."**

[Figure]

Figure R1. The total hydrometeors content of z=35 (~300hPa) at T=10min, unit: g/kg, (a) F00 experiment, (b) G21 experiment.

[Figure]

Figure R2. The water vapor tendency (WVT) at the 35 th model level (~300hPa) from (a,c) cloud scheme and (b,d) convection scheme at integrated time (a,b) 10min and (c,d) 1 hour, in F00 experiment, unit: g/kg/d.

*Line 292: Should "initial time" really be 21Z (i.e., in G21)?*

**Yes, it is 21Z. We added this information, and added "in G21" for "that" to specify the experiment.**

**The sentence in line 292 was modified to: "…the atmosphere of G00 with much weaker supersaturation at initial time than that in G21."**

*Lines 308-309: These findings are definitely in agreement with past studies about the importance of an accurate initial moisture field, at least on the storm-scale. It may benefit the paper and further emphasize the authors' point to add some references to other papers discussing the importance of accurate moisture DA, e.g.:*

*Weygandt, S. S., A. Shapiro, and K. K. Droegemeier, 2002: Retrieval of model initial fields from single-Doppler observations of a supercell thunderstorm. Part II: Thermodynamic retrieval and nu- merical prediction*

*Ge, G., J. Gao, and M. Xue, 2013: Impacts of assimilating measurements of different state variables with a simulated supercell storm and three-dimensional variational method.*

**Many thanks for the information and we highly appreciate it. We added the following sentence to emphasize our point:**

**"It also suggests that we need to pay more attention to the analysis quality of water**

vapor in data assimilation (DA). And this has been also confirmed in previous studies (Weygandt et al. 2002; Ge et al. 2013) that the accurate moisture initial field by DA is an effective way to improve the forecast performance of supercell storm in numerical weather prediction models."

Meanwhile, we added corresponding paper information in reference part.

*Line 329: Do the authors mean the G21 run instead of observations? If not, what observations are being referenced here?*

It is a good question. Yes, we compared the difference of forecast variables in G21 and G00 in section 3.2.1, but did not compare them with the observational dataset. In this part, we mainly focus on the differences caused by not retaining the cloud field information in G00 and we want to check if there is a systematic impact on GRAPES-GFS. When compared with the observational data, the difference changes involve many factors (dynamical core, physical processes, DA, even the model's inherent factor), which are beyond the content of this article. In the further, we will adopt more cycle experiment results to comprehensively evaluate its impact on the forecast bias.

*Line 352: This sentence is unclear to me as I don't understand what is meant by "same forecasts", although I assume the authors are stating that the conclusions for both Lekima and Krosa are the same and therefore only Lekima will be presented. Please clarify.*

Sorry for the confusion. The understanding of the reviewer is exactly what we would like to deliver. We modified the sentence by following the reviewer's suggestion:
"Since the conclusions for both "Lekima" and "Krosa" are the same, only "Lekima" will be presented in this study."

*Line 355: Can "CCWV" and "TCIW" be made consistent with the axis labels in Fig. 9, of vice versa?*

Yes, we corrected the inconsistencies of "CCWC" and "TCIW" between the text in line 355 and the axis labels in Fig. 9. Meantime, we changed "total content of ice water (TCIW)" to "column cloud ice content (CCIC)" throughout the text, which is consistent with the expression of column cloud water content (CCWC).

*Figures 1, 2, 3: Tick labels are small and hard to read. Please enlarge.*
We replotted Figures 1, 2, 3 and enlarged the tick labels.

*Figure 4: Legend text is too small to read.*
We replotted Figure 4 and enlarged the legend text.

*Figures 1, 2, 3, 4: Please add titles to each subplot of what run, height, time, etc. are being shown in each panel. It is confusing having Figure 1 vary by run in each row, Figure 2 vary by run in each column, etc.*
We replotted Figures 1, 2, 3, 4 according to the reviewer's suggestions. For Figure 1, we think it vary by run in each row.

*Figure 7a: Are the legend labels switched here? As per the discussion, shouldn't g21_cwp be higher than g00_cwp?*

**Yes,the legend labels were switched in Fig. 7a. We made the correction.**

*Technical Corrections:*

*Line 32, 37, elsewhere: Change "Besides" to "In addition"*

*Line 43: Change "reasonability" to "representativeness"*

*Line 53: "model"→"modeling"*

*Line 56: "could" → "can"*

*Line 91: "widely-used" → "widely used"*

*Line 115: "difference" → "differencing"*

*Line 128: Should "1.2" be "2.2" here?*

*Line 143: "outputted" → "output"*

*Line 145: "operational solution" → "the operational setup"*

*Line 182: "in the" → "due to"*

*Lines 185-186: "this level" → "these levels"*

*Line 187, 198, 202, 208, elsewhere: "convections" → "convection"*

*Line 244: Remove "analysis"*

*Line 251: "no matter the" → "regardless of whether the"*

*Line 273: "lead time" → "forecast time"*

*Line 277: "at time 1 hour after" → "at 1 hour into"*

*Line 302: "can reflect" → "reflects"*

*Line 303: Remove "relatively"*

*Line 305: "It" → "This"*

*Line 312: "in the operation" → "operationally"*

*Line 313: "less THC" → "decreased THC"*

*Line314: Remove "situation"*

*Line 315, 351: "typhoon track" → "track"; "landed on" → "made landfall in"*

*Line 327: "cloud" → "clouds"*

*Line 334: "moments" → "times"*

*Line 335: "two" → "four"*

*Line 351: "continued to develop on ocean"   → "remained offshore"; remove "from offshore areas"*

*Line 360: "strengthening" → "increase"*

*Line 368: "get" → "gets"; remove "of them"*

*Line 371: "an alternation of" → "alternating"*

*Line 374, 381: "While" → "In contrast,"*

*Line 382: Should "G20" here by "G21"?*

*Line 388: Should "Lichma" by Lekima?*

*Line 398: "All the three different experiments" → "All three experiments"*

*Line 407: "unobvious" → "not obvious"*

*Line 418: "analysis" → "analysis of"*

**We highly appreciate the detailed comments from the reviewer, and agree with the**

reviewer for all the technical corrections. Based on these comments, we made corresponding changes, which have helped us improve the article's clarity. We give special responses to the questions for Lines 128, 382, and 388.

(1) Yes, "1.2" should be "2.2" in line 128.

(2) Yes, "G20" should be "G00" in line 382.

(3) Yes, "Lichma" should "Lekima" in line 388.

---

## Author Comment (AC2) · 5 Nov 2020

Response to Reviewer 2:

We thank the anonymous referee for his/her valuable comments and suggestions that have helped us improve the paper quality. Our detailed responses (**Bold**) to the reviewer's questions and comments (*Italic*) are listed below.

**Anonymous Referee #2:**

*Overview:*

*The manuscript investigates the influence of spin-up and restart in a global weather forecast system GRAPES. Such a topic is important, as careful handling of those technical issues can greatly improve the accuracy of weather prediction. By comparing different spin-up and restart methods, the authors gain important knowledge of the forecasting system, such as that the GRAPES with its own analysis field performs better than the one using NCAR final reanalysis (FNL) data for the cold start in the spin-up. The paper contains useful information for model development and usage. I recommend its publication with GMD, pending on some minor comments below.*

**We highly appreciate the reviewer's positive evaluation about this study. We also thank the reviewer for the valuable and detailed comments and suggestions which have helped us improve the paper quality.**

*About the experiment setup. To better illustrate the differences between three experiments, which are of great importance to this paper, can the authors use a schematic plot to show how the three runs were performed and what are the key input data. Also, it should be explicitly stated in the Section 1.2, why those three experiments were conducted, or in other words, what we expect to learn by comparing them. The three-hour lag confuses me a little bit.*

**Fig. R1 shows the 4D-Var cycle assimilation system and the experiment setup. In fact, we have listed the experimental settings in Table 1. Since the contents of hydrometeor are not analyzed and updated in the 4D-Var system and the cloud information simulated by the G21 experiment is not retained during the restart in the G00 experiment, the input variables of the three tests are the same, we no longer specified the input variables for the three experiments.**

**To more clearly state why three experiments were conducted in the current GRAPES_GFS operational system, we rearranged the fourth paragraph in the Introduction section and added the following sentences: "Then another question is what advantages the new 4D-VAR assimilation analysis fields have in spin-up process compared with the cold start simulation with FNL.", "Actually, for numerical weather prediction model's users (especially forecasters), they are usually accustomed to referring the forecast productions of model staring to integrate from 00 UTC or 12 UTC (or more time, for example 06 UTC, 18 UTC). " , and "The reasons for the unretained cloud-field variables were mainly based on the following considerations: the hydrometeor contents are very small amount relative to water vapor and they can be quickly created in the spin-up process when the model restarts. Moreover, this treatment can save storage space and input/output (IO) time. However, its impacts on the spin-up process and model forecast performance have not yet been carefully analyzed and evaluated. Therefore, we need to fully diagnose and analyze**

**the necessity of the repetition of GRAPES_GFS spin-up during the re-integration, and the impact of the lost cloud-field information on the later forecast. ".**

[Figure]

Figure R1. The schematic diagram of 4D-Var cycle assimilation system and experiment setup in this paper.

*It is unclear to me what is the current protocol for spin-up and restart strategies used by CMA that uses GRAPES_GFS to conduct the daily weather prediction as well as the extreme weather prediction like typhoons. According to this study, is there any modification needed on the protocol?*

**It's a good question. In current protocol of operational GRAPES_GFS it still adopts analysis fields from the 4D-Var system as the model's initial field at 09UTC/21UTC and then restarts after 3hr-integration without the information of cloud fields, that is to say, same to the G21 experiment in this paper. Our research results show that it could lead to systematic biases for height, temperature and precipitation fields as well as typhoon track if the restart of the model does not include the information of cloud fields. We have told the results to the managers of NWPC/CMA, which attracted their considerable attentions. As we all know, the adjustment of numerical weather predication protocol has strict specifications, which needs to carry out parallel experiments for a period of time and evaluate the results before its operation application. The parallel experiment has been listed in the operational testing plan. If there are further results, we will be willing to share with you.**

*Is the total grid number of cloud (TGNC) related with the total cloud fraction? The latter is a more common term. Also, 1.0 e-4 g kg-1 threshold of cloud sounds an arbitrary choice. Are the results sensitive to this threshold definition?*

**It's a good question. TGNC is related with the total content of all hydrometeors (THC, that is to say, THC=cloud water + raindrop + cloud ice + snow + graupel). We define the grid with cloud when its THC is greater than 1.0 e-4 g kg-1 according to our results. We have tried three thresholds (THC= 1.0 e-5, 1.0 e-4, and 1.0 e-3, respectively) to compare the spinup time, and the results we got are basically the same. In other words, teh results seem not very sensitive to the threshold definition. Meantime, to clarify this part more clearly, we added the description for the definition of equilibrium in section 3.1.3 as**

follows "Note that the statistical equilibrium state is defined when the difference of TGNC with respect to the 24-hour integration is insignificant (the difference is less than 20% of TGNC at 24-hour). " combing with the comments of the first reviewer.

*Fig. 11a, why g00 and g21 are identical before 42 Hour and then become different abruptly?*

The forecasted track errors before 42 Hour simulated by G00 and G21 experiments are not identical. Actually, their tracks are very close before 42 Hour, which are shown in Fig. R2. For the abrupt track difference after 42 Hour, it was caused by the continuous accumulation of the direct cloud-radiation processes, temperature difference, and even re-undergone spin-up process in the typhoon cloud area and their transmissions to the typhoon eye through dynamic processes with the integration. As stated in this paper, the restarted model (G00 experiment) with lost cloud-field information in initial field needs to re-undergo a spin-up process and causes systemic biases of cloud, temperature and geopotential height and precipitation fields at the model early forecast. These biases mainly exist in areas with clouds. For a typhoon, the differences of temperature and geopotential height of the G00 experiment initially exist in the cloud belt around the typhoon eye compared with G21 experiment. With the model integration, the peripheral system difference gradually affects the typhoon center (track) through the dynamic process. These changes can be confirmed from Fig. R3 and Fig. R4. For example, the large value area of temperature difference between G00 and G21 experiments at the early stage of integration is mainly located in the spiral cloud belt around the typhoon eye and its value can reach 2k, while its value over the typhoon central is only -0.25~0.25K. With the integration of the model, the temperature difference of the typhoon eye gradually becomes larger, and its value reaches 0.5-1K at 50 hours of integration, which is bound to affect the track of the typhoon by dynamic process.

[Figure]

Figure R2. The track of typhoon "KROSA" observed (the black line) and simulated by G21 experiment (the green line) and G00 (the red line) experiment from 0000 UTC on August 9 to 0000 UTC on August 12, 2019.

[Figure]

[Figure]

Figure R3. The temperature of 850 hPa simulated by G00 (the red line) and G21 (the black line) experiments and the difference of G00-G21 (shad) at different integration time, the unit is K.

[Figure]

Figure R4. The geopotential height of 850 hPa simulated by G00 (the red line) and G21 (the black line) experiments and the difference of G00-G21 (shad) at different integration time, the unit is gpm.

---

## Author Comment (AC3) · 5 Nov 2020

We highly appreciate your positive reminders and suggestions to ensure that the experimental dataset, analysis scripts for the results and the model code can meet GMD's requirements for publishing article. Our detailed responses to each question are listed below.

(1)No model source code. The code and data availability section merely states that code is not available due to "the confidentiality requirement". GMD does permit model code to be withheld from publication if this is unavoidable for reasons beyond the control of the authors. Usually this is because the copyright licence of the code does not

[Figure]

permit redistribution. However, the reasons that the authors cannot release the code must be detailed in the code and data availability section. In particular, it is important to state who can get a licence and how.

We agree with the Editor and appreciate this helpful suggestion. We have added the reason that we cannot release the code in the "code and data availability" part, which is "the model code cannot be distributed due to the copyright licence requirement from the Numerical Weather Prediction Center of China Meteorological Administration (NWPC/CMA). If someone wants to use the GRAPES_GFS model or reproduce these experiments in this article, he/she can contact the operational management department of NWPC/CMA via email songzx@cma.gov.cn or phone +86-10-68400477."

(2) Version not identified. Neither the title of the manuscript nor the code and data availability section state precisely which version of GRAPES_GFS was used. This makes it impossible to reproduce the work even if one has a licence.

Sorry for our mistake. We adopted the version 2.3.1 of GRAPES_GFS in this research and now have specified it in the manuscript: 1) Lines 17 and 18: "Then, the characteristics of spin-ups in the version 2.3.1 of GRAPES (Global/Regional Assimilation and Prediction System) global forecast system (GRAPES_GFS2.3.1) under different initial fields are compared and analyzed." 2) Line 127: "On 1 July 2018, the GRAPES global 4-dimensional variational (4D-Var) data assimilation system has been in operation (Zhang et al., 2019), which is called version 2.3.1 of GRAPES_GFS (abbreviated as GRAPES_GFS_2.3.1). And the GRAPES_GFS_2.3.1 version was adopted in this research." 3) We also specified the model version in the rest of the manuscript.

(3)Model data is not on a persistent public repository. The model data appears to be on a cloud storage provided by Baidu. This lacks the persistence, non-revocability and persistent identifiers required for a journal publication. The data should instead be stored in a properly persistent archive with a persistent identifier such as a DOI. I note in this regard that the authors are from national laboratories so I would expect such

facilities to be available to them.

The Editor proposed a good question. However, our lab doesn't provide a data-sharing platform for individuals, so I had to buy cloud storage of Baidu to share these dataset. I have purchased five years of service and will be able to renew my account in the future. Meanwhile, I stored the data in our high-performance computer storage devices. These could guarantee long-term sharing of the experimental data.

(4)No configuration, run, or data processing scripts. The configuration files, run scripts and any data processing or analysis scripts used to produce the results presented in the manuscript need to be publicly and persistently archived, and cited from the code and data availability section. As a guide, every file the user would need to reproduce the manuscript should accessible.

The experimental configuration file and the data processing and analysis scripts used to produce the results presented in the manuscript have been uploaded to the website. 1) The directory of "data_processing_scripts" includes all the analysis scripts for plotting the results. 2) The directory of "model_configure" includes the configure files and run scripts for the GRAPE_GFS.

---

## Author Response (AR2)

**Response to Editor Olivier Marti:**

We highly appreciate your positive reminders and suggestions to make corresponding changes in the manuscript about Fig. 11. Our detailed response is listed below.

*Comment: Thank you for your responses to the reviewers and the revised manuscript. They address all the points raised by the reviewers in a convincing manner. I still have a little concern. You made a long response to reviewer #2 about Fig. 11. But you didn't change the manuscript accordingly. Could put a phrase in Fig. 11 caption to explain of the trajectories are slightly different even though the difference is to small to be seen on the plot ? I really feel that it would better if it's in the legend.*

Reply: We agree with the editor. We added the corresponding descriptions to explain the differences in the trajectories shown in Figure 11 at Lines 411-414 in the manuscirpt: **"The abrupt track difference after 42th hour is most likely caused by the continuous accumulation of the direct cloud-radiation process and the systematic temperature bias in the typhoon peripheral cloud area during the re-undergone spin-up of G00 experiment, along with their impacts on the typhoon eye (track) through dynamic processes with the model integration.".**